# Epigallocatechin-3-Gallate (EGCG): New Therapeutic Perspectives for Neuroprotection, Aging, and Neuroinflammation for the Modern Age

**DOI:** 10.3390/biom12030371

**Published:** 2022-02-25

**Authors:** Ashley Payne, Samuel Nahashon, Equar Taka, Getinet M. Adinew, Karam F. A. Soliman

**Affiliations:** 1Division of Pharmaceutical Sciences, College of Pharmacy and Pharmaceutical Sciences, Institute of Public Health, Florida A&M University, Tallahassee, FL 32307, USA; ashley1.payne@famu.edu (A.P.); equar.taka@famu.edu (E.T.); getinet1.mequanint@famu.edu (G.M.A.); 2Department of Agricultural and Environmental Sciences, College of Agriculture, Tennessee State University, Nashville, TN 37209, USA; snashashon@tnstate.edu

**Keywords:** Alzheimer’s disease (AD), inflammation, oxidative stress, microglia, aging and epigallocatechin-3-gallate (EGCG)

## Abstract

Alzheimer’s and Parkinson’s diseases are the two most common forms of neurodegenerative diseases. The exact etiology of these disorders is not well known; however, environmental, molecular, and genetic influences play a major role in the pathogenesis of these diseases. Using Alzheimer’s disease (AD) as the archetype, the pathological findings include the aggregation of Amyloid Beta (Aβ) peptides, mitochondrial dysfunction, synaptic degradation caused by inflammation, elevated reactive oxygen species (ROS), and cerebrovascular dysregulation. This review highlights the neuroinflammatory and neuroprotective role of epigallocatechin-3-gallate (EGCG): the medicinal component of green tea, a known nutraceutical that has shown promise in modulating AD progression due to its antioxidant, anti-inflammatory, and anti-aging abilities. This report also re-examines the current literature and provides innovative approaches for EGCG to be used as a preventive measure to alleviate AD and other neurodegenerative disorders.

## 1. Introduction

The global burden of neurodegenerative ailments increases due to aging populations, longer lifespan, and changing environments. The world health organization (WHO) has shown that neurological diseases constitute 12% of total global mortality and contribute to 16.8% of total deaths in lower-middle and 13% in high-income countries [1]. By 2030, neurological diseases, AD, and other dementias will contribute 38% of global disability, as calculated as years of life lost due to disability [1]. The World Health Organization reported that in 2015 the international cost of dementia was projected to be USD 818 billion (approximately 1.1% of global GDP) [2]. Neurodegenerative diseases, such as Huntington’s disease (HD), Parkinson’s disease (PD), multiple sclerosis (MS), amyotrophic lateral sclerosis (ALS), and Alzheimer’s disease (AD), are caused by different etiological factors, including environmental, molecular, and genetic factors. Various hallmark determinants of neuro-degenerative diseases’ pathogenesis include inflammation, elevated reactive oxygen species (ROS), aging, and epigenetic instability.

Alzheimer’s disease was first identified in 1906 by a German physician and pathologist Alois Alzheimer who was able to identify plaque formation and neurofibrillary tangles in his 51-year-old female patient. Still, it was not until 70 years later that a better understanding of its causes, processes, and properties was ascertained [3]. The nervous system is highly susceptible to neurodegeneration because it is anatomically isolated, requiring substantial energy to function and its nonregenerative capability. Finally, neurotransmission is the primary source of biological communication [3]. Glial cells, primarily astrocytes and microglia, are the main progenitors of neuro-degeneration due to their role in providing immunoregulatory neuroprotection.

Alzheimer’s diseases display the common pathology of specific protein misfolding, which partially unfold under physiological conditions and present an intra-molecular cross-beta sheet conformation that produces toxic insoluble fibrillar structures [4]. This toxicity can originate due to the following: (1) loss of function as the conversion to a misfolded conformation diminishes the pool of functional protein; (2) disordered cellular membranes; (3) dysfunctional mitochondrial function and creation of ROS; and (4) burdened regulatory networks of protein homeostasis [4]. Protein misfolding and aggregation due to aging has been shown to enhance inflammatory mediators, i.e., proinflammatory cytokines/chemokines, nitric oxide, and microglia involved in inflammation resulting in neurodegeneration. Phytochemicals have shown promise in modulating the conditions of protein misfolding toxicity due to their roles as antioxidants, anti-inflammatory, and iron chelators [5]. One of these phytochemicals is EGCG, which inhibits amyloid aggregation by direct interaction with misfolded proteins. It has also been postulated that EGCG can ameliorate amyloid fibril transformation [4,6]. EGCG’s potential as a neuroprotective phytochemical may be potentially beneficial in alleviating the neuroimmunological mechanisms associated with AD progression. Furthermore, EGCG may be able to alleviate AD progression by rejuvenating immunosenescence and reducing metabolic stress (i.e., diabetes and obesity) that can exacerbate AD progression.

## 2. Neurodegeneration: Examination of Two Prevalent Neuroregressive Disorders

Neurodegenerative disease pathology consists of many molecular factors, as shown in Figure 1: (a) peculiar protein dynamics coupled to faulty protein degradation and aggregation, (b) oxidative stress and free radical development, (c) deficient bioenergetics and mitochondrial dysfunction, and (d) subjection to metal toxicity and pesticides [7]. Neuronal decline increases with age and is considerable in elderly individuals, manifested in the many neuropathologies.

### 2.1. Parkinson’s Disease (PD)

The most extensively studied neurodegenerative ailments have been PD and AD. PD is widespread in aging populations (individuals over age 65) and is second only to AD in its incidence. It involves neuronal death in the substantia nigra, which results in striatal dopamine deficiency and Lewy bodies: intracellular debris containing fragments of Alpha-synuclein (αS; α-synuclein) [8]. The symptoms of PD are resting tremor, rigidity, bradykinesia, gait difficulty, and postural instability; nonmotor symptoms include depression, anxiety, emotional changes, cognitive impairment, sleep difficulty, and olfactory dysfunction [9]. Research has shown that neurodegeneration in the substantia nigra region is due to the reduction of neuromelanin, as shown by the loss of dark grey pigment in this area [10]. Physiological changes of dopaminergic neurons due to aging result in a decrease in dopamine levels and loss of expression of the dopaminergic phenotype [11]. Other non-dopaminergic neurons interconnected with dopaminergic neurons are also affected, such as cholinergic neurons of the pedunculopontine nucleus and glutamatergic neurons of the intralaminar thalamic nucleus.

PD’s association with microglia and neuroinflammation comprises overactive microglia that can also release countless amounts of cytokines and chemokines, leading to neurodegeneration aging [12,13]. New scientific investigation shows that αS serves a dual role as a pathological progenitor in its role in protein aggregation, but also aids in normal immune function [13,14,15]. PD, similar to AD, shows the consequences of the aging brain [16]. To date, epigenetic alterations of DNA methylation caused by aging also contribute to neurodegeneration, which is displayed in both AD and PD pathologies [16].

The pathogenesis of Parkinson’s disease involves not only aging and genetics, but also environmental factors such as cigarette smoking and exposure to pesticides [11]. These have been linked to causing mitochondrial dysfunction resulting in the PD pathology. Neurotoxins in the brain can result in oxidative stress and neurotransmission stagnation, which can have detrimental effects in the basal ganglia [17]. The production of reactive oxygen species (ROS) occurs when hydroxyl radicals are generated from hydrogen peroxide under the Fenton Haber Weiss reaction. This reaction may cause oxidative stress and neurotoxicity to various cellular compartments, primarily affecting the mitochondria via hydroxyl radicals, which respond to deoxyribonucleic acid (DNA), membrane lipids, and proteins of the cell, contributing to their eventual dysfunction. The latest treatments available are Levodopa therapy that, if combined with carbidopa, shows promise as a peripheral dopa decarboxylase inhibitor. The other therapy is the use of dopamine agonists, catechol-o-methyl-transferase (COMT) inhibitors, anticholinergics, and monoamine oxidase inhibitors (Mao-B). Subthalamic stimulation is recommended only for the clinically and biologically fit [18].

### 2.2. Alzheimer’s Disease (AD)

AD is the most prevalent neurodegenerative disease that affects more than 35 million worldwide (5.5 million in the U.S.). The prominent risk factor is aging, which doubles every five years after 65 and an increase in predominance of the disease (13.2–16 million cases in the United States) by mid-century [19]. African Americans (AA), Latinos, and Hispanics are more susceptible to cognitive impairment and delayed onset compared to Caucasians due to the prevalence of cardio/cerebrovascular disorders, genetics, socioeconomics, and racial/ethnic discrimination [20].

AD is a genetically complex disorder, and more than 100 rare mutations have been described [21]. AD has genetic determinants in the form of the early onset (EOAD) or familial AD (FAD), which involves the mutation of one of three genes, amyloid precursor protein (APP), presenilin 1 (PSEN1), or presenilin 2 (PSEN2). These genetic alterations allow for the early development of AD (30–50 years of age). Although it is a rare (affecting less than 1% of all AD cases) autosomal dominant inherited form of AD, it has enhanced scientific knowledge about AD development and pathology (particularly neurofibrillary tangles and tau hyperphosphorylation). APP is the most widely studied due to its involvement with amyloid-beta, but the presenilins are significant fragments of the atypical aspartyl complexes that function in the cleavage of γ-secretase to APP, and PSEN1 is responsible for EOAD [22]. The other form is sporadic or late-onset AD (LOAD), which is nondominant and involves the allele Apolipoprotein ε4 (APOE4), which accounts for 95% of AD incidents [23,24,25]. APOE ε4 has been linked to a greater risk of AD in patients with Down Syndrome, traumatic brain injury, and stroke. It has also been linked with metabolic risk due to its responsibility in cholesterol/triglyceride metabolism [22]. APOE adheres to definite receptors, i.e., LDL receptor-related protein 1 (Lrp1) and very-low-density lipoprotein receptor (VLDLR), that expel chylomicron and VLDL residue from circulation, allowing for normal triglyceride-rich lipoprotein catabolism [26,27].

Many hypotheses have been proposed for AD’s origin—the cholinergic, Aβ, and inflammation hypothesis. The amyloid cascade hypothesis suggests that neurodegeneration arises due to a sequence of events activated by the abnormal processing of the amyloid-beta precursor protein (APP) [8,28]. The cholinergic hypothesis suggests that cholinergic instability leads to AD pathology. Meanwhile, the neuronal cytoskeletal degeneration hypothesis indicates that cytoskeletal changes are the cause. Furthermore, the metal theory postulates that metal ions are responsible for AD genesis. Moreover, inflammation has been related to microglial activation, reactive astrocytes, and heightened cytokine expression, leading to AD [29].

The theory behind the inflammation link to AD is that inflammatory processes of one system are synonymous with those of another system. The inflammatory regulators seem to be enhanced in areas of high AD pathology (such as the frontal neocortex and limbic cortex). Furthermore, stimulated microglia and reactive astrocytes aggregate near fibrillar plaques. Chronically stimulated microglia release chemokines and lead to a sequence of events of damaging cytokines, i.e., interleukins IL-1, IL-6, and tumor necrosis factor-α (TNF-α). Microglia contain receptors for higher-level glycation end products, which bind Aβ resulting in boosting the generation of cytokines, glutamate, and nitric oxide (NO). Microglia also dislodge Alpha-1-antichymotrypsin (ACT), alpha-2-macroglobulin (A2M), and c-reactive protein (CRP), which can intensify AD [30].

One of the main clinical attributes of AD is protein aggregation in the form of extracellular amyloid-beta (Aβ) plaques and intracellular neurofibrillary tangles (NFT). Patients exhibit prolonged cognitive decline, which presents itself as lapses of time and forgetfulness. It can continue to affect speech, higher-order reasoning, visuospatial abilities, and behavioral changes. In the later stages, the patient loses total ability to perform rudimentary activities such as eating, toileting, and dressing, resulting in complete reliance upon a caregiver. Current diagnosis is based on patient history evaluation that assesses mental state via a cognitive function test and physical examination that examines vascular and neurological signs. Current drug treatment for AD has been limited due to a dearth of effective biomarkers, complicated clinical trial designs, indefinite clinical measurements, hindered differentiation from other dementias, hampered focus on treating the underlying origin of CNS disease, and deficiency of predictive animal models [31].

In light of these facts, the therapeutic strategies are mostly symptomatic treatments with non-steroidal anti-inflammatory drugs (NSAIDs) shown to decrease the risk of AD and delay progression, but only in prospective observational studies by halting cyclooxygenase 2 (COX-2) or the prostaglandin E2 receptor, activating phagocytosis by microglia, triggering peroxisome proliferator-activated receptor gamma (PPAR-γ), and selective decrease of Aβ42 [19]. The Food and Drug Administration (FDA) approved acetylcholinesterase inhibitors (AChEI) (rivastigmine and donepezil) and NMDA receptor antagonist (Memantine) drugs have been utilized to alleviate the cholinergic and glutamatergic disintegration/excitotoxicity, respectively, leading to synaptic disruption caused by the AD pathology [32,33,34]. Present therapeutics are disease-modifying agents mainly focused on the Aβ propagation, but they have failed in clinical trials [35]. Other targeted strategies are sirtuins (SIRT), which have recently been associated with age-related diseases, and caspases, which are involved in apoptosis and autophagy [36]. Recently, more studies have focused on preventing the risk factors and co-morbidities, in particular, diet and obesity [37], cerebrovascular (the brain renin-angiotensin system (RAS) [38]), and energy dysregulation [39].

### 2.3. Protein Aggregation in AD: The Role of Amyloid Beta (Aβ)

Extensive research has linked APP with AD pathogenesis [40]. The adhesion to APP occurs in two unique ways: α-secretase, a proteolytic enzyme that triggers the nonamyloidogenic pathway, which leads to APP joining within the Aβ region, resulting in Aβ generation. Adherence of APP by α-secretase generates α-secreted APP (sAPPα) and the leftover fragment C terminal (α-CTF). The other route involves the BACE-catalyzed β-attachment of APP promoting Aβ formation and the amyloidogenic pathway. BACE cleavage produces β-secreted APP (sAPPβ) and the c-terminal fragment β-CTF of APP. The c-terminal remnants (α and β CTF) are substrates for γ-secretase, a universal multimeric protease. The binding of α-CTF to γ-secretase yields a shortened nonamyloidogenic peptide (p3). However, β-CTF adherence promotes Aβ propagation. Aβ is evidence of the main proteinous remnant of amyloid plaques in the AD brain [41].

The APP intracellular domain (AICD) is generated in both the amyloidogenic and nonamyloidogenic pathways via the joining of γ-secretase. Most amyloid-beta remnants are Aβ40 in length; however, the elongated Aβ 42 variant is the most pathogenic. A possible inference is that the continuous assembly of Aβ oligomers causes synaptic damage culminating in neurodegeneration [3]. The latest research has shown that Aβ can activate microglia and promote a chronic inflammatory state resulting in unregulated pro-inflammatory cytokine release, leading to neurodegeneration [42,43]. Alasmari et al. [40] demonstrated that proinflammatory cytokines could mediate Aβ neurotoxicity via APP protein homeostasis/metabolism. Aβ has also been linked to causing iron dyshomeostasis, stimulating oxidative stress in microglia leading to neurodegeneration [44,45].

### 2.4. Tau Protein: Neurofibrillary Tangles and Aggregation in AD

Tau is related to a family of microtubule-associated proteins (MAPs) that reside mainly in neurons, function to support microtubules, and aid in proper cellular trafficking and cytoskeleton scaffolds for cells. Tau protein contributes to the formation of neurofibrillary tangles due to impaired oxidative phosphorylation or apoptotic function [46]. Tau becomes pathological due to hyperphosphorylation that occurs due to unstable kinase and phosphatases acting on the tau protein [47]. The phosphorylation of tau has been associated with N methyl D aspartate receptor (NMDAR) at postsynaptic locations, which may be related to the synaptic instability that occurs in PD [48]. The mechanism of filament aggregation that occurs in tau is still being investigated. The process of fibrillogenesis (similar process to Aβ fibrillogenesis) has been postulated as one of the mechanisms. First, protein dimerization occurs through the arrangement of disulfide bridges or through ionic bonds; then, a nucleation phase follows, in which the dimers self-assemble to generate oligomers; lastly, the elongation phase occurs in which the oligomers thicken. Tau autosomal dominant mutations in microtuble associated protein (MAPT), the gene encoding for tau, were found to cause frontotemporal dementia with parkinsonism (FTPD-17 mutation) [49].

## 3. Role of Inflammation

Current research has led to neuroinflammation as being a prominent factor in the genesis of neurodegeneration. The innate immune response to viral infection or traumatic brain injury induces glial cell activation, i.e., microglia and astrocytes, essential for immunoregulative central nervous system (CNS) homeostasis [50]. This regulation is carried out in a series of processes conjunctionally between the microglia and astrocytes, leading to neurodegeneration at various molecular intervals involving cell death mediation, synaptic remodeling, as well as immune-signaling regulation, i.e., (a) apoptosis, (b) necroptosis, (c) autophagy, (d) retrograde degeneration, (e) Wallerian degeneration, (f) demyelination, and (g) astrogliopathy [51].

The complement system serves as a pivotal pathway of regulating the interplay of acute vs. chronic neuroinflammation within the CNS microenvironment and links to understanding CNS immunoregulation [52,53,54]. Neuroinflammation is normally beneficial for the proper control of external stressors. Still, it can become detrimental when the immune response is prolonged (or chronic) due to immunosenescent aging, which leads to immune signaling dysregulation culminating in neurodegenerative pathogenesis [55,56]. De Oliveira et al. [57] discuss the disruption of natural energy (mitochondrial) homeostasis by the mitochondrial damage-associated molecular patterns (mDAMPs), which can lead to receptor-mediated (NOD-like receptor (NLR) and TLR) cascade of inflammatory-immunological response. Continued research is focused on the relationship between chronic inflammation and AD.

### 3.1. Microglia and the Debilitating Effects Caused by Aging

Microglia are specialized macrophages of the CNS. Microglia function in surveillance of the CNS microenvironment through six divisions: (1) proliferation, (2) morphological transformation, (3) motility and migration, (4) Intercellular communication, (5) phagocytosis, and (6) proteostasis. Microglia differ from other glial cells, such as astrocytes, by their hematopoietic origin and main responder to pathogen infection and injury [58]. Microglia contain many macrophage-associated markers, i.e., CD11b and CD14. These cells are activated by either the classical or alternative complement system; classically, microglia become activated due to pathogen invasion and initiate the major histocompatibility complex II (MHC-II), which recruits T cells and other proinflammatory mediators. They also contain many receptors that function in surveillance, such as the pattern recognition receptors (Toll-like receptors (TLRs)), retinoic acid-inducible gene-I-like receptors (RIG-1 like receptors), and nucleotide-binding oligomerization domain-like receptors (NOD-like receptors). Protein recognition receptors (PPRs) sense pathogen recognition molecular patterns (PAMPs), whereas TLRs detect damage-associated molecular patterns (dAMPs) [59,60].

In AD, progression seems to render microglial cells less efficient in degrading these dense aggregates [61]. Microglia also contain many other receptors, such as ion channels, that aid in microglial recognition of disease-associated molecular patterns (DAMPs) expelled from damaged or necrotic cells and serve a role in the clearance of debris and stimulation of tissue repair after CNS injury, such as purinergic receptor and receptor for advanced glycation end products (RAGE). The purinergic P2 purinoreceptors are categorized into two subgroups: ionotropic receptors (P2X receptors), which create ion channels that are opened by adenosine triphosphate (ATP) and metabotropic receptors (P2Y receptors), which bind purines or pyrimidines and initiate downstream signaling events through G-protein coupled receptors (GPCRs) [62]. Nucleoside triphosphates (NTPs) expelled from injured cells can attach to P2X or Y receptors and stimulate the proinflammatory transcription mediators via the transcription factor, nuclear factor kappa B (NF-κB), and activator protein I (AP1). Microglia have also been shown to crosstalk with astrocytes and intensify the immune response. Microglia is a crucial factor in neurovascular maturation and is involved in angiogenesis [63]. This role has allowed for extensive research in its association with various diseases related to the brain, such as ischemic stroke, brain tumors, and neurodegenerative disorders.

Aging causes a gradual reduction in physiological functions and behavioral capabilities, which is displayed at many phases of the organism, especially in the CNS [64]. Microglia experience many age-related transformations that advance to promoting a chronic mild inflammatory environment, i.e., elevated production of inflammatory cytokines and ROS. Research has shown that the normal brain mass atrophies 2–3% per decade after 50 [64]. Using magnetic resonance technology (MRT) and voxel-based morphometry, atrophies were present mainly in the gray and white matter volume changes in the prefrontal, parietal, and temporal areas. Furthermore, complex learning functions are diminished with age. At the cellular level, aging leads to shrinking telomeres and initiation of tumor suppressor genes, compiling DNA damage, oxidative stress, and imbalance of pro- and inhibitory inflammatory cytokine activity. Mosher and Coray [65] made the important statement, “The hallmarks of microglial aging may also sometimes be characterized as dysfunctional and even hyper-reactive responses.” It has been postulated that cumulative stimulation in response to systemic infections over the course of a lifetime leads to systemic inflammation, causes the cells to be hyperactivated, and aids in neurodegeneration. In addition, an increase in the engineering of ROS via permeability of the blood-brain barrier (BBB) leads to deterioration of neurons and other brain cells and heightened systemic inflammation [65].

Research has shown that microglia proliferation and self-renewal increases with age. Still, it is believed that this progressive assembly of microglia is a compensatory mechanism to sustain the necessary level of protection from the overall microglial population as the individual cells become less capable with age [65]. Morphological changes in AD show that microglia have short, thick, and poorly ramified processes [65]. Microglial motility is reduced with age and display diminished reactions to exogenous ATP and laser-induced focal tissue injuries. Aged microglia display overexpression of many pro-inflammatory markers, such as MHCII, antigens, CD11b/CR3, and CD14. It has also been shown that normal mediators of microglial activation and inflammatory responses are curtailed due to aging [66]. Microglia normally modulate the phagocytotic processes involved in the immune system. Still, aging diminishes these processes as evidenced in dysregulated proteostasis pathways that involve a disruption in chaperone-mediated protein folding and stability, protein trafficking, protein breakdown, autophagy, and compiling of proteins in AD [65].

### 3.2. Microglia and Tau

Senescence of microglia manifests itself as morphological alterations, cytokine expression, phagocytosis, stimulation, and density [67]. Ongoing inquiry shows that microglia may be responsible for the synaptic and non-synaptic circulation of pathological tau. Microglia envelop both soluble and insoluble states of tau. The envelopment of tau allows for either tau degradation or release in exocytosing microvesicles known as exosomes in which elevated levels are situated in the CSF and blood of AD patients [68]. Further postulation is that dephosphorylated tau contributes to microglial neurotoxicity [69].

## 4. Neuroinflammatory Mediators

Cognitive disruption due to cytokine/chemokine regulated associations between CNS neuronal bodies has been thoroughly researched [70]. AD pathology leads to the overexpression of these mediators resulting in enhanced plaque formation and neuronal cell deterioration [70]. Inflammatory regulators involved in AD are as follows: (1) tumor necrosis factor-alpha (TNF-α), (2) interleukin (IL)-6, (3) interferon-gamma (IFN)-γ, (4) inducible protein-10 (CXCL10), (5) monocyte chemoattractant protein 1 (MCP-1), and (6) C-X-C motif ligand (CXCL-8). Interleukin 6 (IL-6) regulates numerous immune reactions affecting CNS cell development and cellular diversity by cooperating with precise soluble or membrane-bound receptors [30]. IL-6 has been shown to have a myriad of inflammatory regulative properties [71], but it can also stimulate acute phase proteins leading to enhanced vascular permeability, lymphocyte provocation, and antibody creation. IL-6 activation by nuclear factor kappa B (NF-κB) modulates vascular inflammation [72]. TNF-α is another proinflammatory cytokine that has been linked to many neurological disorders. Mice studies have displayed enhanced neuronal loss after lesioning due to the lack of the TNF receptor compared to normal TNF receptor adequate mice. Additional studies have shown that TNF-α instigates the presence of protective molecules, such as manganese superoxide dismutase (MnSOD) [73,74,75]. New investigations have shown TGF-βs raise APP expression in cultured astrocytes and microglia and apolipoprotein-E (ApoE) formation in brain slice culture [30].

Chemokines of the CXC or alpha (α) subfamily consist of positionally conserved cysteine residues at the amino terminus divided by a single amino acid allowing for chemotaxis of neutrophils and endothelial cells (IL-8). Chemokines direct biological responses via seven-transmembrane G-protein coupled cell surface receptors (GPCRs). CXC Motif ligand 12 is involved in neurogenesis and mobilizes neurological bodies to sites of lesions in the CNS [76]. Macrophage migration inhibitory factor (MIF) is a pleiotropic proinflammatory cytokine that enhances the creation of other inflammatory cytokines and is localized in microglia. Research has demonstrated that MIF attaches to Aβ in AD-compromised areas [76]. Macrophage inflammatory protein 1 (MIP-1α) expression is in numerous neurological and immunological areas such as astrocytes, microglia, and T cells. Research demonstrated that the MIP-1α/C-C chemokine receptor type 5 (CCR5) signaling pathway is responsible for stimulated glial cell aggregation, inflammatory reactions, and synaptic/cognitive dysregulation [76]. Vascular inflammation has been implicated as a progenitor of AD in the form of rheumatoid arthritis (RH), atherosclerosis, and cardiovascular disease (CVD). Vascular inflammation may be instrumental in elevating a myriad of proinflammatory mediators acting via cerebral microvessels such as TNF-α and IL-6 [77].

### 4.1. Oxidative Stress

Oxidative stress is a condition that is elevated in the brain with aging, produced by an imbalance in the redox state, involving the generation of excess ROS or the dysfunction of the antioxidant system. Due to the brain’s excessive metabolic rate and decreased capacity for cellular regeneration, it is highly susceptible to ROS [78]. The brain also contains many phospholipids, which are rich in polyunsaturated fatty acids (PUFAs), such as docosahexaenoic acid (DHA) and arachidonic acid (AA). Research has shown that as free radical production rises, the PUFA content in the brain gradually decreases [79]. The lipid hydroperoxides are partially unstable and can automatically degrade into various products, such as malondialdehyde (MDA), 4-Hydroxynonenal (4-HNE), ketones, epoxides, and hydrocarbons in the presence of iron. In the AD brain, there is an elevated level of protein carbonyl. Reactions of various ROS and reactive nitrogen species (RNS) with tyrosine lead to the production of 3-nitrotyrosine and dityrosine [79]. DNA oxidation can result in creating 8-Oxo-2′-deoxyguanosine (8-OHdG), which is highly expressed in mitochondrial DNA of AD patients [79]. In all variations of ROS generation, molecular oxygen must be triggered, and the cellular system has evolved many metalloenzymes that facilitate ROS generation upon the interaction of redox metals with O_2_ utilizing various catalytic pathways [80].

The mitochondrial electron transport chain expends almost 98% of molecular oxygen at the cytochrome oxidase complex, and the remaining oxygen is reduced to hydrogen peroxide and superoxide radicals. During normal metabolism and various functions of the oxygen radical superoxide anion ^−^(O_2_^−^•) and the non-radical oxidant hydrogen peroxide (H_2_O_2_), and hypochlorous acid (HOCl or HClO) are created. When the production of (O_2_^−^) and (H_2_O_2_) becomes unrestrained, they can result in tissue damage that often involves the generation of the highly reactive hydroxyl radical (OH·) and other oxidant molecules in the presence of catalytic iron or copper ions [81]. The brain has abundant peroxidation-susceptible lipid cells and is an organ with a high indulgence for oxygen. Moreover, the cerebrospinal fluid cannot bind expelled iron ions. Consequently, oxidative stress in nervous tissue may seriously damage the brain via several interconnected mechanisms, such as increased intracellular free Ca^2+^, the release of excitatory amino acids, and neurotoxicity [82].

Other progenitors of oxidative stress are RNS, NO, and peroxynitrite (ONOO−), which can be highly reactive with proteins, lipids, nucleic acid, and other molecules resulting in modifying the structure and functionalities and having detrimental effects on the brain. Cells with an accumulation of oxidized products, i.e., aldehydes, protein carbonyls, and base adducts from DNA oxidation, can be seriously transformed. The large ROS formation elevated by the electron transport system within the mitochondria under stressful situations and aging comprises a risk for AD development [81].

NO not only contributes to oxidative stress but also functions in inflammation [83]. NO is an inflammatory mediator in various joint, gut, and lung diseases, i.e., rheumatoid arthritis [84]. NO association with the immune system and glial cells has garnered interest in mediating neurodegenerative diseases, such as AD and PD. Recent research has displayed NO’s role in vascular dementia [85], in which similar inflammatory molecular mechanisms can be used in examining AD pathology.

### 4.2. Nitric Oxide

NO is a multifaceted mediator molecule that is expressed highly in the neurons [86,87]. The enzyme nitric oxide synthase (NOS) was isolated in 1990 in the rat brain (neuronal nNOS or NOS-1), as well as in macrophages (inducible iNOS or NOS-2) and endothelial cells (eNOS or N0S-3) [88]. The enzymatic reactions of iNOS are regulated transcriptionally by cytokines, whereas post-translational modifications via Ca^2+^/calmodulin control the activity of eNOS or nNOS [88]. Normally NO is steadily expelled in small increments and functions in vasodilation. Many different immune system cells (leukocytes, mast cells, platelets, and macrophages) produce and respond to NO [83,86]. NO halts platelet and leukocyte binding to the endothelium, which diminishes pro-inflammatory processes [86]. Aging and inflammation increase the generation of the ROS/RNS superoxide anion (O_2_^−^•) that can react with NO to form peroxynitrite (ONOO-), which interferes with mitochondrial respiration and causes NO dyshomeostasis leading to mitochondrial dysfunction and neuronal death [88,89]. It has been shown that aging causes decreased eNOS-generated NO production, leading to increased cardiovascular risks, which can involve reducing the antioxidant activity, superoxide overproduction, and modifications of NOS enzyme/expression activity. AD may be correlated with vascular degeneration in the central nervous acting via NO signaling dysregulation.

Evidence shows that NO-mediated neurotoxicity in AD is caused by microglial cell stimulation, promoting INOS-regulated NO discharge [89]. Furthermore, pro-inflammatory cytokine TNF-α induces NO assembly. Immunoreactive components such as macrophages can produce superoxide through NADPH oxidase [87]. Microglia, glia, and neurons release high amounts of glutamate [89]. NO can halt respiration and cause glutamate release from synaptosomes and neurons, hindering mitochondrial respiration and reversing glutamate uptake or initiating vascular exocytosis [89].

NO has been shown to cause neuronal death via energy depletion-induced necrosis, oxidative phosphorylation, or excitotoxicity [83]. The mechanisms postulated to be involved, according to the paper by Brown and Bal Price [87], are as follows: (1) stimulation of poly ADP ribose polymerase (PARP) followed by nicotinamide adenosine diphosphate (NAD^+^) and ATP reduction, (2) promotion of apoptosis by weakly understood reactions, (3) glutamate release, and (4) halting of mitochondrial respiration. Research has shown that cerebral blood flow diminishes with aging, promoting endothelial cell dysfunction regarding depressed NO discharge [87]. NF-κB is postulated to be involved due to its inflammatory regulatory properties; eNOS suppresses its activation via proinflammatory cytokines [87]. Microvascolopathy may arise due to the reduced NO release and NF-κB stimulation [87]. NO at reduced concentrations suppresses cytochrome oxidase in contention with oxygen and may serve as a mediator of cellular energy metabolism [88].

### 4.3. Autophagy

Autophagy is an evolutionarily preserved mechanism to sustain neuronal homeostasis during cellular development and maturation. It is maintained by lysosomal action, which controls nutrient recycling in starvation, neurotransmitter release, synaptic restructuring, and pruning during the growth of axons and dendrites [90]. To date, there are three major types of autophagy in mammals: (1) Macroautophagy, which refers to the isolation of cytoplasmic materials into double or multi-membrane vesicles called autophagosomes. The process involves developed autophagosomal transport using microtubules in which cellular byproducts are delivered to degradative compartments; meanwhile, during this operation, lysosomes create autolysosomes. (2) Microautophagy involves the distinct engulfment of the cytosolic substance by invagination of the lysosomal membrane. (3) Chaperone-mediated autophagy (CMA), which consists of proteins having a pentapeptide pattern (KFERQ-like sequence) that are identified by the cytosolic chaperone heat shock cognate protein of 70 kDa (hsc70) and its co-chaperones distribute them to the cell surface of lysosomes [91]. Autophagy plays a pivotal role in cell survival by safeguarding stressful situations, i.e., ER and metabolic stress. Selective autophagy allows better organelle regulation, limiting the dysfunctional compartment, and pathogen disposal by uniting the ubiquitin-proteasome system (UPS) and autophagic apparatus. As efficient as this process may be, conditions caused by overabundance and destabilization can lead to cell death [92,93,94].

Aging and inflammation act as precursors to autophagic dysregulation and bolster cell death by exacerbating the cellular burden misfolded and damaged cellular deposits and diminishing cellular degradative capacity cells and other modifications of intracellular proteolytic systems [95]. It has been demonstrated that autophagy modulates critical elements of the immune system, namely, macrophages, lymphocytes, and neutrophils [96]. Autophagic proteins (Beclin1 or ATG16L1) can mediate proinflammatory cytokine production in macrophages and other inflammatory regulators. It can also, in turn, control cytokine production itself, acting through Il-1β. Countless research in various mammalian organisms (rodents and humans) has shown that aging contributes to oxidative stress, DNA damage, and telomere shortening resulting in impaired autophagy, promoting AD genesis [97,98].

Recent research has shown that in AD, mitophagy (a form of selective autophagy that involves the mitochondria) dysregulation leads to the aggregation of mitochondrial debris and destabilization of mitochondrial integrity leading to cell death [93,97]. The thoroughly researched process of the phosphatase and tensin homolog (PTEN)-induced the kinase 1 (PINK1)-Parkin pathway depicts the decline of the mitochondrial membrane potential, which increases PINK1 on the outer membrane of mitochondria (OMM) to enlist and switch on Parkin, a ubiquitin ligase, via phosphorylation of ubiquitin. Parkin ubiquitinates numerous OMM proteins, which stimulate the UPS to degenerate these ubiquitinated OMM proteins, resulting in the mobilization of the autophagy machinery to instigate the engulfment of impaired mitochondria by phagophore or secluded membranes and thus create mitophagosomes reserved for removal via the lysosomal system [99]. Evidence of oxidative stress within the AD brain include (1) lipid peroxidation, in which lipid byproducts, i.e., 4-hydroxy-2,3-nonenal (HNE); (2) protein oxidation in the form of 3-nitrotyrosine and protein carbonyls were observed in the hippocampus of AD patients; and 3) DNA/RNA damage leads to disrupted DNA repair mechanisms and elevated DNA mutations exhibited within the hippocampus and cerebral cortex [100].

## 5. Inflammatory Signaling Pathways Involved in Neurodegeneration

Normal characteristics of aging in the brain are decreased neurogenesis, elevated synaptic damage, increased metabolic stress, cognitive decline, and immunosenescence. Aging is also linked to systemic inflammation, increased blood-brain barrier (BBB) permeability, and dysregulated glial cell signaling, influencing chronic inflammation [101]. Normal glial signaling regulating the inflammatory response is the NF-kB, mitogen-activated protein kinase (MAPK), and phosphoinositide-3-kinase/protein kinase B/mechanistic target of rapamycin (PI3K/AKT/mTOR) pathways. The NF-κB pathway has been linked with inflammation related to aging. NF-κB can be stimulated by ROS, cytokines, inflammatory agents, growth factors, and synaptic transmission (primarily glutamate). In mice, it was shown that microglia stimulation induces a highly conserved transcriptional signature with aging, characterized by NF-κB expression and neuronal death. In rats, elevated NF-κB expression resulted in the generation of neurodegenerative proinflammatory enzymes: cyclooxygenase 2 (COX-2) and inducible nitric oxide synthase (iNOS). Diabetes has displayed an increased risk of AD by 50% by influencing Aβ pathology by NF-κB upregulation and independent overexpression of beta-site amyloid precursor protein cleaving enzyme 1 (BACE1). Research on primate and rodent models shows that diabetes and obesity drive overexpression of NF-κB in the hypothalamus by promoting a feedback loop of hypertension, overnutrition, and diminished insulin sensitivity [102].

MAPKs consist of serine/threonine kinases triggered via multiple protein kinases in response to extracellular stimuli by dual phosphorylation at conserved threonine and tyrosine residues [103,104]. The extracellular signal-regulated kinase (ERK) pathway is a mediator of neuronal function. ERK1/2 activation occurs by various neurotransmitters and neuropeptides through induction of ligand-gated ion channels or GPCRS; ERK1/2 primarily regulates cell proliferation and may regulate calcium signaling [103]. The c-Jun N-terminal kinase (JNK) pathway has been shown to promote neurodegeneration or neuroinflammation via the provocation of potentially deleterious stimuli such as hypoxia, free radicals, and ROS. The p38 mitogen-activated protein kinase (p38) pathway can be stimulated by various growth factors (platelet-derived growth factor (PDGF), insulin-like growth factor (IGF), and vascular endothelial growth factor (VEGF)), pro-inflammatory cytokines, and oxidative stress resulting in multiple cellular responses such as inflammation, apoptosis, cell growth, and differentiation [103]. IL-6 stimulates Janus kinase (JAK), which activates signal transducer and activator of transcription (STAT), which provoke the regulation of inflammatory response.

The phosphoinositide-3-kinase/protein kinase B/mechanistic target of the rapamycin (PI3K-Akt-mTOR) signaling pathway regulates many cellular metabolism and energy homeostatic processes, such as proliferation and survival, which can involve the immune system and CNS. It mediates neuronal physiological conditions, such as learning, memory, and neuroprotection. PI3K is activated by phosphorylation due to the coupling of various growth factors, such as epidermal growth factor (EGF) and signaling proteins, i.e., (TLR) ligands bind to receptor tyrosine kinase, which allows for the phosphorylation of phosphatidylinositol 3,4-bisphosphate (PIP2) to PI (3,4,5)-triphosphate (PIP3) triphosphate, which stimulates serine-threonine kinase AKT controlled by phosphoinositide-dependent kinase 1 (PDK1), which can trigger mTOR activation. Current research has shown the relevance of this pathway to aging and, coupled with AD, can alter normal AKT and mTOR activity since both are controlled by oxidative stress, decreased nutrient concentration, and innate immune signals. Further, AD upregulates this pathway resulting in neurodegeneration. The PI3K-AKT-mTOR signaling pathway is linked to insulin regulation (particularly cerebral insulin), reduced in aging individuals making them more susceptible to AD [105,106].

## 6. Contributing Determinants to the Incidence of AD: Health Disparities, Sex, and Gender

Health inequality involves differences in quality healthcare due to living conditions, race/ethnicity, socioeconomic status, pollutant exposure, diet, comorbidities, gender, and family health history [107]. Other noteworthy factors are cognitive reserve, genetic differences in pathological resilience, and educational level. There is a dearth of research into health inequality effects on minorities, i.e., African Americans (AAs), Latinos, and Hispanics, related to AD [20,108]. A research article by McDonough [109] introduces the Weathering hypothesis deduced by Geronimus [110], which states “the cumulative impact of those mentioned above social, physical, and economic adversities faced by AAs lead to early health deterioration and advanced biological aging,” which can be used to evaluate social stressor effects on minority neurological health.

Powell et al. [111] evaluated community effects involved in AD, a cross-sectional investigation of autopsy samples from 447 decedents living in a disadvantaged neighborhood at the time of death. They found that it was correlated with a heightened risk of Alzheimer’s disease neuropathology when adjusted for age, sex, and year of death. Another relevant study was conducted by Saadi et al. [112], who showed that AAs and Hispanics were less likely (30% to 40%) to see an outpatient neurologist than Caucasians, respectively. Furthermore, AAs were prone to being cared for in an emergency room, having frequent hospital stays, and paying more for inpatient care than Caucasians. Other research has illustrated how social stressors can exacerbate the risks for AD neuropathy [112,113].

A research area that has not garnered enough attention is the sex and gender differences observed in neurodegenerative disorders. Men and women have different outcomes concerning a neurodegenerative disease arising from socioeconomic status, level of education, genetics, and hormonal differences [114,115,116]. Current studies show that sex differences exist in a cognitive decline concerning AD [117,118]. Bloomberg et al. [119] evaluated cognitive decline and performance correlated to aging and education and found that women had stronger memory and lesser memory decline than men regardless of education level. However, education did play a factor in the fluency tests, in which women that were higher educated and in the later cohort had higher scores than the men. Still, there were no sex differences in fluency decline [119].

Androgens (testosterone) and estrogens may serve to prevent and treat neurodegenerative diseases due to their many anti-inflammatory properties, such as halting proinflammatory cytokine production, protection from oxidative stress, and modulation of Aβ production [120,121]. Several investigations show that estrogen’s neurological roles in regulating the effects of the trophic factors in the brain heighten the cerebral blood flow, restrain atrophy of cholinergic neurons, and adjust BBB glucose transporter expression and membrane translocation [122,123]. Aging is liable to cause a loss of sex hormones (androgens and estrogens), which induce muscle loss, muscle weakness, diminished functional performance, and reduced life span [124,125]. Postmenopausal women (age 65 and over) are more susceptible to AD than men [122]. Immunological sex differences exist in microglia, as shown in microglial transcriptomic and proteomic studies that change with age [126,127].

## 7. Preventive Measures for Neurodegenerative Disease: Nutraceutical and Phytochemical Use in Neurodegenerative Disease

Regarding AD, there have been no new FDA-approved drugs since 2003 and no authorized disease-modified treatment (DMT), although there have been many extensive clinical trials [32,33,128]. Considering this fact, the advent of nutraceuticals has led to a possible new form of control of this disease. Nutraceuticals are a broadly defined term that comprises many products that arise from the food industry, herbal and dietary supplement trade, pharmaceutical, and hybrid agribusiness/nutrition enterprises that may have potential medicinal value in treating disease [96]. Nutraceuticals have gained tremendous interest due to their long history of use and providing a natural alternative to the current pharmaceutical drugs in the market [96]. Nutraceuticals have been shown to have anti-inflammatory, antioxidative, and anti-aging properties. The anti-inflammatory/antioxidative processes carried out by nutraceuticals are as follows: (a) hinder NFκB activation, (b) inhibiting overexpression of proinflammatory cytokines, (c) downregulation of the overexpression of calcium/calmodulin (CAM) and enzymes, (d) block enzymatic action of COX-2 and iNOS, (e) arrest ROS enzymatic production, and (f) elevate ROS scavenging [129].

Phytochemicals are nutraceutical plant chemicals with either defensive or disease protective roles [130]. Some common phytochemicals are green tea polyphenols, phytoestrogens, anthocyanidins, carotenoids, and terpenoids. Phytochemicals have been shown to have many antioxidative and anti-inflammatory properties, as evidenced using berberine, which has been found to stimulate the phosphoinositide-3-kinase/protein kinase B/ nuclear factor erythroid 2 related factors 2 (PI3K/Akt/nrf2) pathway to negate ROS generation. The use of phytochemicals in autophagy has been displayed in cancer research. As stated previously, autophagy is a catabolic process in which damaged or impaired molecules and organelles are degraded by lysosomes for removal from the body.

One example of a phytochemical is quercetin (3,3′,4′,5,7-pentahydroxyflavone), which resides in many fruits and vegetables, such as apples and berries, and onions have shown autophagic induction in Schwann cells by alleviating the cell damage generated by elevated glucose [131]. It also diminishes lipopolysaccharide (LPS)-induced nitric oxide release from a mouse neuroglia cell line [131]. Further anti-inflammatory actions were found in luteolin (5,7,3′,4′-tetrahyderoxyflavone), which is found in celery, parsley, green pepper, chamomile tea, and perilla leaf, and decreases iNOS and COX-2, and thus downregulates inflammatory mediators, formation of NO, and prostaglandin E2 in LPS-activated BV2 microglial cells. It also enhanced neuron viability and decreased apoptosis in rat primary hippocampal neurons stimulated with lipopolysaccharide (LPS) administered at 20 mM [132].

### Green Tea and Its Derivatives: Therapeutic Actions Related to Inflammation and Neurodegeneration

Tea is the second most frequently drank beverage in the world after water [133,134]. Tea has been consumed both socially and habitually since 3000 B.C. *Camellia sinensis* (L.) (*C. Sinensis* L.), which is a member of the Theaceae family (evergreen plants), is a resident of China that later spread to India, Japan, Europe, Russia, and the New World (Americas) in the 17th century. Green, Oolong, and black tea all originate from the same plant species, *C. Sinensis* L, but are dissimilar in appearance, organoleptic taste, chemical content, and flavor [135]. The chemical constituents of green tea leaves consist of polyphenols (catechins and flavonoids), alkaloids (caffeine, theobromine, and theophylline), volatile oils, polysaccharides, amino acids, lipids, vitamins, inorganic components, aluminum, fluorine, and manganese). Green tea consists of six main catechin compounds, catechin, gallocatechin, epicatechin (EC), epigallocatechin (EGC), epicatechin gallate (ECG), and EGCG, which is the most active component. Two derivatives that are being researched thoroughly are gallic acid (3,4,5-trihydroxybenzoic acid) and (−) epicatechin gallate (EC).

GA is a plentiful phenolic compound found throughout the plant kingdom. It furnishes and improves the anti-inflammatory, antioxidative, and neuroprotective attributes of EGCG [136,137]. An analysis performed by Mori et al. [138] revealed that GA administration rescinded impaired learning and memory in a mutant human amyloid β-protein precursor/presenilin 1 (APP/PS1) transgenic AD mouse model. It also reduced cerebral amyloidosis and increased nonamyloidogenic APP processing. Brain inflammation, gliosis, and oxidative stress were quelled. GA was able to raise α and β secretase response, impede neuroinflammation, and balance brain oxidative stress in a pre-clinical mouse AD model [138]. Moreover, in the same study, GA displayed the ability to elevate disintegrin and metalloproteinase domain-containing protein 10 (ADAM10), proprotein convertase furin, and initiate ADAM10, which inhibits BACE1 activity. In an investigation by Andrade et al. [139], GA exhibited anti-amyloidogenic properties (in which in vitro neuronal membranes), which modulate the GA-induced Aβ fibril disaggregation, that could be associated with the moderate affinity of the compound for the lipid membrane. GA can cooperate with α-synuclein to halt amyloid fibrillogenesis in PD progression [140,141,142].

Liu et al. [143] demonstrated that GA diminished LPS-induced elevation in heme oxygenase-1 level (a redox-regulated protein) and α-synuclein accumulation, proposing that GA is capable of inhibiting LPS-induced oxidative stress and protein conjugation. In addition, GA hinders LPS-induced caspase 3 activations (a biomarker of programmed cell death), and LPS stimulates increases in receptor-interacting protein kinase (RIPK)-1 and RIPK-3 levels (biomarkers of necroptosis). This suggests that GA inhibits LPS-induced apoptosis and necroptosis in the nigrostriatal dopaminergic system of rat brain co-incubation of GA attenuating LPS-induced increases in iNOS mRNA and iNOS protein expression in the treated BV-2 cells, as well as NO production in the culture medium. Most importantly, GA protects against mitochondrial dysfunction, DNA damage, and apoptosis that occurs in AD, as evidenced in a rat model in which it altered a multitude of antioxidant enzymes such as SOD, glutathione peroxidase (GPx), catalase, glutathione-s-transferase (GST), and glutathione (GSH) content [137]. GA provides antioxidative protection against a chemically induced hypoxia/reoxygenation model of cerebral damage by inhibiting the mitochondrial ROS accumulation and simultaneously increasing oxidative phosphorylation and ATP synthesis [137].

EC is a green tea derivative, but it can also be found in berries, apples, grapes, peanuts, and cocoa tea. EC has the general chemical structure of other flavonoids of the C6-C3-C6 configuration. It consists of two aromatic rings linked by a heterocycle formed by an additional three-carbon chain and one oxygen atom. Current research is focused on metabolic stress disorders that are associated with neurodegenerative disease development. One, in particular, is obesity, which is caused by a consistent imbalance of energy intake and expenditure characterized as abnormal or excessive fat build up that disrupts the normal adipose tissue surveillance and energy regulation. This leads to adipose tissue inflammation, which can lead to system-wide comorbidities, such as insulin resistance, T2D, cardiometabolic disorders, and advances the aging process, resulting in heightened risk for age-related diseases, i.e., AD and PD [144,145]. EC can quell GI inflammation via its antioxidant properties (decrease NADPH upregulation and heightened oxidant production) and anti-inflammatory functions by reducing the stimulation of inflammatory signaling (ERK1/2 and NF-κB) [146]. EC demonstrated the ability to mediate lipid regulation as exhibited in dyslipidemias, which enhances the risk of atherosclerosis [146]. EC has been shown to alleviate insulin resistance in in vivo animal studies by reducing glucose and insulin levels, refining insulin sensitivity, and improving glucose metabolism [146]. Finally, EC has been shown to improve cognitive impairment, but also to elevate brain-derived neurotrophic factor (BDNF), which is a neurotropin involved in learning and memory. EC may have neuroprotective abilities by enhancing cerebral blood flow and attenuating endotoxin-mediated inflammation caused by inflammatory mediator aberrant transport [146].

## 8. EGCG Synthesis, Structure, and Therapeutic Action

EGCG is an ortho-benzoyl benzopyran byproduct [147], comprised of four rings marked A, B, C, and D (Figure 2). A and C embody the benzopyran ring, which has a phenyl group at C2 and a gallate group at C3. The B ring has positional 3,4,5-trihydroxyl groups, and the D ring galloyl moiety is configured as an ester at C3. The medicinal properties of green tea are due to EGC esterification with gallic acid (galloylation), allowing for the antioxidative mechanisms of green tea correlated to EGCG [148].

Green tea polyphenol acting by way of EGCG’s unique structure allows for highly efficient antioxidative properties. EGCG counteracted superoxide anion and hydrogen peroxide and blocked ROS-induced DNA damage. EGCG is a peroxynitrite scavenger reducing the nitration of tyrosine and a forager of free radical byproducts, such as hypochlorite and peroxyl radicals. It can also act as a chelator of iron and other metals via its phenolic groups, allowing binding to inactive forms of Fe(III), Cu, Cd, and Pb, leading to reduced amounts of free forms of these metals, thus promoting ROS reactions [149]. In addition, green tea polyphenol antioxidative functions are displayed by enhancing the activity of glutathione peroxidase (GPX) and superoxide dismutase (SOD), and the scavenging rate is much stronger than vitamins C and E [150].

### EGCG Bioavailability and Modifications

Despite EGCG’s many medicinal properties, it possesses many shortfalls when it comes to bioavailability (the extent and rate at which the bioactive compound enters the systemic circulation and penetrates the site of action). EGCG has inadequate systemic absorption following oral administration, including reduced pharmacokinetics, limited biodistribution, first-pass metabolism, and lowered accumulation in related tissues of the body or low targeting efficacy [151]. The poor intestinal absorption is due to oxidative decomposition under high temperature, neutral or weak alkali conditions, and the breakdown rate increases with the elevation of ambient temperature and oxidation concentration [152]. Furthermore, intestinal pH may be the prime factor affecting the stability of oral EGCG [152]. The retention time and permeability of intestinal EGCG affect the absorption of EGCG; the deficient permeability is caused by the inadequate intestinal transport caused by passive diffusion and active outflow [152]. After oral administration, EGCG is metabolized into GA and EGC in the small intestinal microbial system, while EGCG further degrades into 5-(3,5-dihydroxyphenyl)—valerolactone (EGC-M5) in the large intestine [153]. EGCG has a higher BBB permeability, which may be due to its hydrophobicity (the less polar the molecules, the better the absorption of brain tissue) [152]. Some of the current modifications that have been proposed in modifying EGCG’s bioavailability are nanostructure-based drug delivery systems, which utilize encapsulation materials, i.e., lipids, proteins, and carbohydrates as carriers. One example is the study by Liu et al. [154] in which a palmitate version of EGCG was isolated and determined as 40-0-palmitoyl EGCG. It had better stability and durability to digestive enzymes (α-amylase). Current studies show the lipid nano-carriers to be the most effective over the other nanoparticles due to their high stability and biocompatibility, controlled release properties, low-cost production methods, and easy scalability [155,156].

## 9. Epigallocatechin-3-Gallate (EGCG) Therapeutic Action in AD and PD

Current therapeutics seek to limit the production of amyloidogenic peptides, heighten the natural state of amyloidogenic proteins, increase the clearance rates of misfolded proteins, and specifically halt the self-assembly process [157,158,159]. Green tea polyphenols have exhibited many disease-alleviating properties, specifically EGCG, which has shown many medicinal attributes, particularly neuroprotective (as evidenced in Figure 3). Animal research has shown that EGCG has anti-aging abilities due to its capability to act as a free radical scavenger [160]. EGCG has been shown to affect many potential targets related to AD. It can protect against beta-amyloid-induced neurotoxicity in cultured hippocampal neurons. It has also illustrated a capability to mediate the processing of APP, through protein kinase C (PKC) activation, to the nonamyloidogenic soluble amyloid precursor protein (sAPP), thus preventing the formation of the neurotoxic Aβ. Furthermore, EGCG has been shown to halt the beta-secretase enzyme (BACE1), which is tasked with processing sAPP to Aβ [65]. EGCG was shown to halt the neurotoxicity caused by Aβ by stimulating the glycogen synthase kinase 3 (GSK3) coupled with restriction of cAbl/FE65, which is a cytoplasmic tyrosine kinase needed for proper progression of the nervous system and nuclear translocation [161].

In PD, EGCG’s medicinal properties have been exhibited in its neuro preventative abilities, as evidenced in an in vitro study in rotenone (a pesticide that induces Parkinson’s symptoms similar to humans) in rats. This investigation showed that EGCG reduced the measurable molecular impairments of PD, such as lipid peroxidation, oxidative stress (NO biomarker), and other neuroinflammatory and apoptotic markers for PD [162]. Tau phosphorylation may be prevented by EGCG [163] via ionic bond strength [164]. A solutions’ ionic strength (through NaCl) can modify the rate of aggregation via changing the electrostatic attractions between protein molecules. This can alter the conformation and morphology of the fibrils that coalesce during aggregation [164]. In another study, it was shown that EGCG could stop human tau phosphorylation and interact with tau using various biochemical and bioanalytical experiments, such as isothermal titration calorimetry and thioflavin S (ThS) fluorescence assay [165].

## 10. Autophagic Role of EGCG

Several research investigations are examining the mechanisms of EGCG mediation of autophagy using various in vitro disease cell model systems. Kim et al. [166] showed that EGCG, at a low concentration of 10 µM, could induce autophagy and autophagic flux in endothelial cells that aid in the degeneration of lipid droplets via the calcium (Ca^2+^)/calmodulin-dependent protein kinase β (CaMKKβ)/5′AMP-activated protein kinase (AMPK)/CaMKKβ/AMPK-dependent mechanism. Zhao et al. [147] detected that EGCG-prompted autophagy initiates the degradation of the α-fetal protein (AFP, which functions in cellular differentiation and maturation) aggregates in the hepatocarcinoma (HepG2) cell line. Meng et al. [167] found that EGCG upregulated the levels of autophagic proteins Atg5, Atg7, LC3 II/I, and the Atg5–Atg12 complex in human vascular endothelial cells (HUVECs), while downregulating apoptosis-related proteins. It also halted the PI3K-AKT-mTOR signaling pathway, thereby partially promoting EGCG-induced autophagy. Holczer et al. [168] observed that in the human embryonic kidney 293 transfected (HEK293T) cell line, EGCG was able to prolong cell viability by stimulating autophagy via the mTOR pathway; it was also able to inhibit apoptosis by upregulating autophagic survival. In the absence of growth arrest and DNA damage-inducible protein (GADD34) in the presence of GADD34 (critical autophagy regulating gene) inhibitors (guanabenz or siGADD34). Lastly, an in vivo study was performed by Khalil et al. [169] in which EGCG’s methylation inhibitor properties were able to inhibit DNA methyltransferase 2 (DNMT2) (which is correlated to increased levels of methylated autophagy molecules ATg5 and LC3B) in C57BL/6-aged (62–64 weeks old) mice.

These actions by EGCG show that the induction of autophagy may promote neurorescue and its anti-aging functions in AD. A possible mechanism of action is that the induction of autophagy may minimize the protein aggregation caused by Aβ [66]. Moreover, the antioxidative properties of EGCG may hinder the ROS associated with mitochondrial dysfunction and utilize autophagy to act as an ROS scavenger mechanism. Lastly, the promotion of autophagy can reduce the proinflammatory cytokine/chemokine activity caused by chronic neuroinflammation from overactive glial cells. A summary of the proposed scheme is exhibited in Figure 4.

## 11. Other EGCG Therapeutic Applications for Neuroinflammation and Neurorescue in Other Related Pathological Disorders

EGCG’s medicinal characteristics have been explored in multiple cells and animal models involving inflammation, immunomodulatory, and ROS regulation, especially in the contemporary literature [170]. As discussed previously, in the medicinal properties of green tea, the focus has been on the link between the inflammatory effects on the cardiovascular system and its correlation with neurodegenerative diseases. This is evidenced in cigarette smoking, which heightens ROS production and contributes to the upsurge of inflammatory response seen in coronary heart disease and stroke [171]. EGCG displayed the ability to diminish cigarette smoke-stimulated inflammation in human cardiomyocytes utilizing the MAPK and NF-κB signaling routes. Airway inflammation is exhibited in bronchial asthma, a chronic respiratory disorder that is increasing globally due to air pollution and other environmental changes [171]. It was shown that EGCG elevated IL-10, CD4^+^, CD25^+^, and Foxp3^+^ Treg cells and expression of Foxp3 mRNA in lung tissue, which all contributed to a reduction in airway inflammation in the female BALB/c mice model [171,172]. T cell dysregulation exacerbates atherosclerosis advancement, which is correlated to congenital heart disease. The integration of T cells into atherosclerotic lesions expel pro-inflammatory cytokines, which promote the expression of MHCII complex antigens on macrophages and vascular smooth muscles. EGCG can halt this proinflammatory expression, thus negating T cell dysregulation [173].

Another example of immunomodulation by EGCG is on the receptor activator of NF-κB (RANKL), which is expressed in osteoblasts, epithelial cells, and initiated T cells. EGCG inhibited caspase-1 activity and reduced transcriptional activity of nuclear factor (NF)-κB by halting inhibitory protein κBα phosphorylation in RANKL-stimulated HMC-1 cells [174]. Moreover, rheumatoid arthritis, a chronic age-related inflammatory disorder, was shown to be elevated by ROS generation, but EGCG has exhibited ROS reductive capabilities in a Wistar mouse model to balance the oxidative-antioxidative system [175]. Lastly, metal toxicity (such as Pb^3+^ and Cd^2+^) contributes to and worsens inflammation and normal energy homeostasis in the cerebro-cardiovascular and other related systems [149]. EGCG has been shown to reduce this toxicity via its antioxidative properties [176,177,178,179].

The brain vasculature, which consists of a complex distribution of arteries, veins, and capillaries, circulates essential substances like oxygen and glucose to the brain while eliminating waste products such as CO_2_ [63]. Proper blood flow is important for optimal brain function [63]. Various research has been focused on the antioxidative properties for the treatment of coronary artery disease, which is correlated with high oxidative stress and dysfunction of the endothelium. Green tea does not reduce blood pressure or plasma lipids, but does inhibit lipid peroxidation, reduce cholesterol levels (in mice), and minimize the development of aortic atherosclerosis in rabbits [160].

Several investigations have shown that cardiovascular dysfunction (hypertension, diabetes, atherosclerosis, and ApoE allele ε4) is linked to AD [180,181,182]. AD-like amyloid-beta deposits have been observed in the neutrophil and within neurons of non-demented patients with heart disease [180,181,182]. Vascular tissue dysfunctions found in patients with AD involve diminished microvascular density, blood vessel fragmentation, atrophy, elevated capillary irregularity, alterations in blood vessel diameter, increased thickness of basement membrane, and collagen accumulation in the basement membrane [180]. It has been postulated that a defective cerebrovascular system could halt the removal of Aβ, resulting in increased concentration in the brain. Furthermore, BBB disruption may allow for plasma proteins and fibrinogen to enter the brain parenchyma, which can induce inflammation and instigate neurodegeneration [183].

Aging alters the integrity of the brain and cognitive ability by intensifying the biomechanical force of the cerebral vasculature and changing vascular remodeling [184]. Chronic inflammation is an inducer of atherosclerosis by stimulating the vascular endothelium, elevating the adhesion of mononuclear cells to the incapacitated endothelial layer, and extravasation into the vessel wall. Green tea polyphenols may alleviate vascular endothelial dysfunction and inflammation by upregulating eNOS levels, elevating VEGF expression, and halting endoplasmic reticulum/oxidative stress. The green tea polyphenol EGCG was observed to constrain angiotensin II and TNF-α-induced hypertrophy by inhibiting ROS-promoted stress in cardiomyocytes [185]. Epidemiological studies have shown that green tea consumption causes a 5-10-fold lower incidence of PD in Asian populations [160]. The interconnectedness between the immune, cardiovascular, and nervous systems are linked to endothelial dysfunction and inflammation, which are factors involved in the pathogenesis of AD that may be alleviated by the medicinal actions of EGCG.

### Role of microRNA (miRNA) in AD: Medicinal Action by EGCG

MicroRNAs (miRNA) are non-coding single-stranded RNAs (usually 22–23 nucleotides in length) that regulate gene expression on the 3′ untranslated region (UTR) of messenger RNA (mRNA) by halting translation or initiating degradation of the marked mRNA [186,187]. miRNAs reside in the nervous system, where they control advanced neuronal processes such as synaptic plasticity. MicroRNAs (miRNAs) can modulate the innate and adaptive immune responses, mainly miR-21, miR-155, miR-125b, and miR-146a, which are drastically up-regulated in neurodegenerative diseases [188]. miR155 is an established pro-inflammatory miRNA for the innate immune response; resveratrol (3,4′,5-trihydroxy-trans-stilbene) can debilitate the up-regulation of this miRNA by LPS in a miR-663 dependent approach [188]. In AD, miRNAs have displayed the ability to direct the activity of APP and BACE1, thus suppressing the production of Aβ as evidenced, presently, in mIR-132 [189,190].

Current research has shown that miRNAs may be able to stimulate TLRs and mediate neuroinflammatory processes by acting on inflammatory cytokines (examples are mIR-9 and mIR 155), which suppress inflammation by regulating downstream proinflammatory activators such as TNF receptor-associated factor 6 (TRAF6). miRNAs may also serve as effective biomarkers due to their appearance in serum, plasma, or cerebrospinal fluid, allowing for degradative protection and stability from environmental insults, as well as their ability to be easily gathered and analyzed via current technology, i.e., next-generation sequencing (NGS) [189]. In terms of EGCG effects on miRNA in neurodegenerative disease, in particular, AD, there is a dearth of knowledge, although the anti-inflammatory effects of EGCG have been shown to increase miRNA expression in chondrocytes and reduce inflammation in osteoarthritis by acting on miR-199a-3p by reducing the stimulation of COX2. The production of prostaglandin E2 (PGE2), as well as interleukin 1, will be inhibited, and its effects on the enzyme ADAM metalloproteinase with thrombospondin type 1 motif (ADAMTSS) [191,192]. Furthermore, EGCG was shown to reduce the prevalence of miRNA in the serum of an APP/PS1 transgenic mouse [193]. Based on this, EGCG may act indirectly on miRNA to reduce age-related neuroinflammation and act as a neuroprotective measure in AD pathology. Although miRNA treatment exhibits promising and innovative approaches to alleviating AD, it also has its shortcomings, such as the association between miRNA and the gene of interest not always being 1:1, which makes gene targeting difficult. Next, natural variations in miRNA expression patterns are to be pondered during experimental design. Single-stranded miRNA show decay kinetics in particular circumstances. Lastly, data interpretation of miRNA expression depends on the detection platform used [187].

## 12. EGCG in Clinical Studies of AD and PD

Most of the examination of the medicinal benefits of EGCG in neurological disease has been performed in vitro and in vivo, as displayed in Table 1. The clinical trials pertaining to EGCG, AD, and PD were acquired from ClinicalTrials.gov and are shown in Table 2. A total of 3 studies were taken from this site, in which two were completed and only one is ongoing. A detailed description is shown in Table 1. In the first study (NCT00951834), participants were elderly adults (aged 60 and over). The inquiry aimed to evaluate EGCG’s anti-protein aggregation properties in AD by preventing the induction of alpha-secretase and the endothelin-converting enzyme, as well as to prevent the aggregation of beta-amyloid to toxic oligomers through the direct binding to the unfolded peptide. There were no results mentioned for this investigation. The next examination (NCT03978052) was recruiting elderly adults and focused on the premise that many adjustable risk factors for AD have been identified in observational studies, which are radical and do not apply any effects through amyloid or tau. This implies that primary prevention studies focusing on risk reduction and lifestyle modification may offer further benefits. The last analysis (NCT00461942) was to ascertain whether EGCG/ECG is effective and safe in the treatment of de novo Parkinson’s disease. This study observed 30-year-old adults, and the results were not mentioned. Unfortunately, due to its poor bioavailability and inconclusive evidence as an effective monotherapy, there is a dearth of clinical data related to autophagy, neuroinflammation, and senescence related to EGCG and AD. Ongoing investigation coupling EGCG with other agents may show promise [194]. As mentioned previously, various pharmaceutical modifications to improve the effectiveness of EGCG are ongoing. The proposal of a new model system to investigate autophagy in neurodegenerative disease may show promise, as discussed by Tzou et al. [195] in the use of drosophila.

## 13. Lipids, Cholesterol Metabolism, and AD: A New Possibility for EGCG?

Lipids have a multitude of functions, i.e., cellular metabolism, structural integrity, and energy modulation. Aging, unregulated food consumption, and reduced physical activity have led to a worldwide epidemic of obesity, insulin resistance, and metabolic conditions leading to diabetes, hyperlipidemia, and hypertension [206,207]. Body fat composition increases with age and is stored primarily in the abdominal region, elevating the susceptibility to cardiovascular disease and diabetes in the elderly [208]. Aging is also linked with a reduction in fat oxidation, which results in fat accumulation. Fat oxidation involves the release of fatty acids from adipose tissue and the aptitude of respiring tissues to oxidize fatty acids. Lipids serve as progenitors for many secondary messengers, i.e., arachidonic acid (AA), docosahexaenoic acid (DHA), and 1,2-diacylglycerol (DAG) [209]. The brain lipid composition comprises sphingolipids, glycerophospholipids, cholesterol ester, and fragments of triglycerides.

Cholesterol is a significant portion of the brain (the brain is a majority cholesterol-rich organ) due to its role as a necessary constituent of cell membranes [210]. It serves as the progenitor of steroid hormones, bile acids, fats, and lipophilic vitamins. Cholesterol mediates proper synaptic plasticity, axonal direction, and synaptic development [211]. It also regulates many physiological brain functions, primarily through its concentration in lipid rafts. Cholesterol also mediates apoptosis (mitochondrial cholesterol) and clearance mechanisms (lysosomal cholesterol) [212]. Cholesterol can be acquired through the diet or endogenous synthesis; therefore, cholesterol homeostasis relies on the regulation of lipoprotein trafficking. Modifying cholesterol metabolism by aging may heighten the risk for metabolic and cardiovascular diseases and neurodegenerative diseases, such as AD. The BBB serves to halt the uptake of lipoproteins from circulation and regulate its clearance via the conversion of cholesterol to its metabolite—24-hydroxycholesterol [150,213].

Cholesterol biosynthesis is a multivarious process involving converting acetyl coenzyme A (Acetyl CoA) to 3-hydroxy-30 methylglutaryl-CoA by hydroxymethylglutaryl-CoA (HMG-CoA) synthase, which is changed to mevalonate by HMG-CoA reductase. A sequence of enzymatic reactions converts mevalonate into 3-isoprenyl pyrophosphate, farnesyl pyrophosphate, squalene, lanosterol, and via a 19-step process to cholesterol [214]. Cholesterol biosynthesis dysregulation has effects on the two significant age and inflammatory intracellular pathways mechanistic target of rapamycin (mTOR) and the NAD^+^-dependent deacetylase silent information regulator proteins (sirtuins) [215].

Statins are drugs that halt cholesterol biotransformation through the competitive inhibition of 3-hydroxy-3-methylglutaryl co-enzyme A reductase (HMGCR), thus preventing the conversion of HMG-CoA into mevalonate [216]. Scientific investigation has shown that statins can also provide neuroprotection by modulating nitric oxide production and eNOS, enabling a reduction in ischemic stroke and reducing the ROS involved in this condition [216]. Furthermore, statins contribute to neurorescue and utilize cholesterol-dependent mechanisms by decreasing interferon-stimulated expression of MHC-II on antigen-presenting cells (APC) [216]. Serum cholesterol levels are instrumental in Aβ promotion in AD, so its mediation by a potential HMGR inhibitor deserves further research and evaluation. A contemporary study demonstrated that statins could alleviate the cognitive deficiency of Sprague-Dawley male rats with experimental AD, lessen the stimulation of microglia and astrocytes, stall apoptosis, and downregulate TLR4, tumor necrosis factor receptor (TNFR)-associated factor 6 TRAF6 expression and mRNA/protein levels of the NF-κB pathway [217].

Aging heightens the plasma levels of low-density lipoprotein cholesterol (LDL-C) within the body, whereas high-density lipoprotein (HDL-C) levels decrease [215]. It has been postulated that oxysterols may be responsible for AD progression as evidenced in their elevated levels in AD patients (such as 27-hydroxycholesterol (27-OHC) and 7-ketocholesterol (7-KC)) (as shown in Figure 5A,B) [210]. Insulin acting via glucose homeostasis in the brain is involved in AD development due to destabilized glucose/cholesterol metabolism [210]. TNF-α has been associated with hyperlipidemia and obesity acting via lipolysis or lipoprotein activity regulation [218,219]. Cytokine activation of microglia can be mediated by a high-fat diet [220].

In the liver, insulin resistance enhances hepatic glucose production and lipogenesis, advancing to hyperglycemia and lipotoxicity-induced pancreatic β-cell dysfunction. The liver also serves as a clearance mechanism in the brain [221]. Zhou et al. [222] displayed how EGCG elevated hepatic autophagy by inducing the generation of autophagosomes, enhanced lysosomal acidification, and triggered autophagic flux in hepatic cells and in vivo as well as lipid clearance. EGCG’s antioxidant, fatty acid, and cholesterol metabolism effects reduced glucolipid metabolism and oxidative stress in type 2 diabetic rats [223]. EGCG was shown to act on the 5′ AMP-activated protein kinase (AMPK) phosphorylation and modify gut microbiota [224]. Lastly, EGCG regulated the lipid metabolism in the acclaimed poultry model for obesity: the boiler chicken [202].

## 14. Future Directions

AD is on pace to be the fastest-growing worldwide epidemic in the next few years [225]. Current research on AD should be focused on advancing our understanding of phytochemical intervention in inflammaging, cholesterol metabolism, microglia-neuronal interactions, epigenetics, neuroprotection, and autophagy to create a more robust alternative therapy to combat AD. The enhancement of EGCG bioavailability and combination immunotherapies may be of scientific interest. Research focused on BBB barrier dysfunction and mitochondrial disruption by environmental toxicants on in vitro models and the administration of green tea catechins such as EC and EGCG. Scientific analysis on health disparities and comorbidities, especially in association with metabolic stress, may be of concern. This review also highlights the need for in vitro studies on green tea catechins (EGCG/EC) coupled with physical activity and calorie restriction in preventing neurodegenerative disease. Lastly, researchers should evaluate preventive measures by green tea catechins on protein aggregation/misfolding and neuroinflammation in concussion and reduction of susceptibility to neurodegenerative pathogenesis.

## 15. Conclusions

This review showcased the therapeutic actions of polyphenols and introduced the medical benefits of green tea catechins. The ameliorative and neuroprotective properties of EGCG were discussed in relation to neuroinflammation, aging, protein aggregation, and autophagy (Figure 6). EGCG was shown to quell neuroinflammation by reducing microglial activation. Aging was discussed as the main factor of heightening neurodegenerative disease development. This was discussed in association with the immunosenescence of microglia. AD and PD were used as the main archetypes of neurodegenerative pathology, and both are rising in significance as global aging populations increase. The protein tau was introduced and discussed for its role in understanding pathological fibrillogenesis. Autophagy, a common research interest in cancer, has gained interest in neurodegenerative disease to understand the dysregulated clearance mechanisms demonstrated in PD and AD. Metabolic stress was examined in relation to the current curative properties of EC, and various anti-oxidative functions of GA were deliberated. The postulated role of EGCG in mediating cholesterol metabolism was introduced. Lastly, a debated topic of health disparities, sex, and gender were included to deal with the challenge of unequal access to curative therapies due to socioeconomic conditions. EGCG remains a promising therapeutic strategy in the battle against neurodegenerative disease.

## Figures and Tables

**Figure 1 biomolecules-12-00371-f001:**
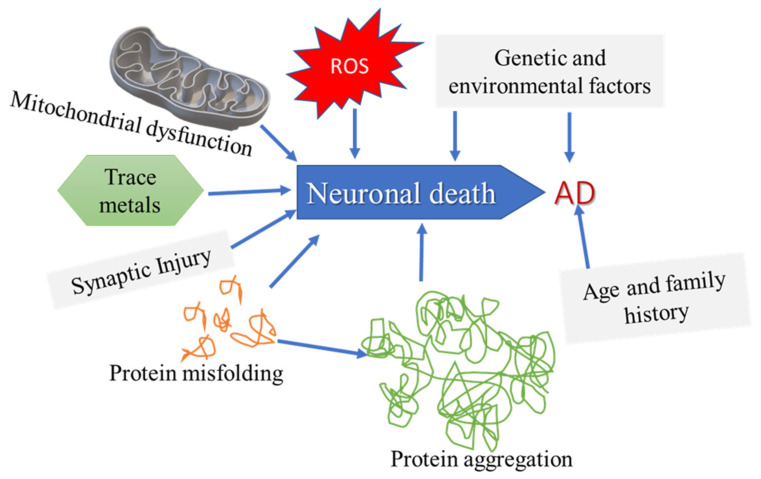
Common etiological factors that might lead to neurodegeneration. The etiological origins of neurodegenerative diseases such as AD are unknown, but the known stimulators are multifactorial, i.e., genetic, environmental, aging, and molecular. AD displays the typical characteristics associated with neurodegeneration: protein misfolding provoking protein aggregation, synaptic dysfunction, metal toxicity, mitochondrial dysfunction, and ROS.

**Figure 2 biomolecules-12-00371-f002:**
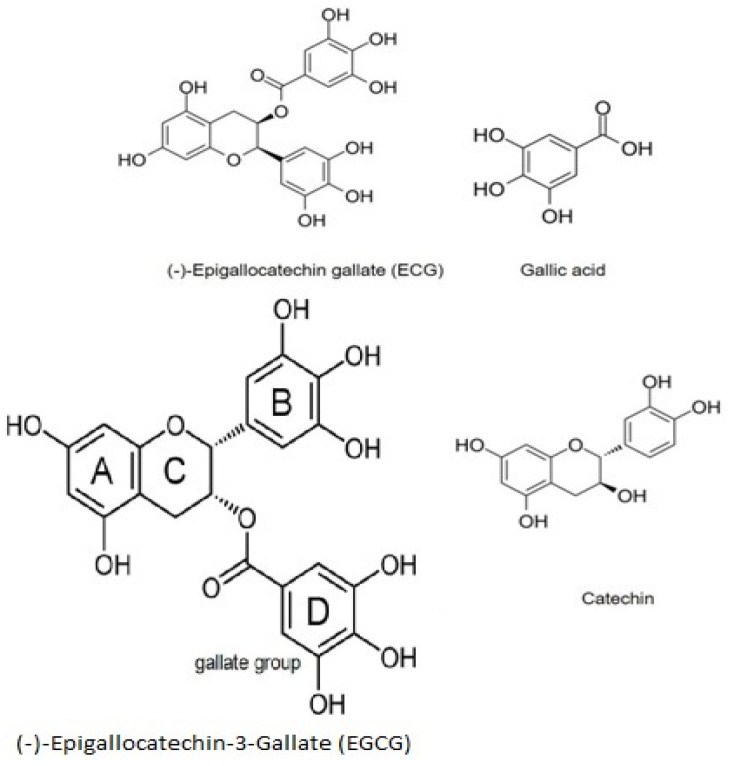
Chemical Structure of EGCG and its progenitors. The chemical makeup of EGCG contributes to its ROS reducing and anti-inflammatory properties. EGCG comprises 4 rings denoted A, B, C, and D. A and C form the benzopyran ring, which has a phenyl group at C2 and a gallate group at C3. The B ring has positional 3, 4,5-trihydroxyl groups, and the D ring galloyl moiety (gallate group) is configured as an ester at C3. The B and D rings have contributed to their ROS deactivating properties. The D ring has been shown to be associated with anti-inflammation and anticancer characteristics.

**Figure 3 biomolecules-12-00371-f003:**
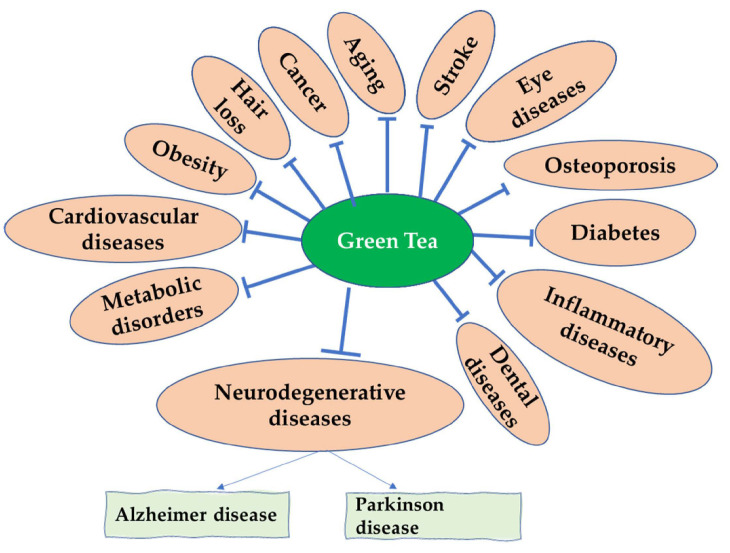
Medicinal Properties of Green Tea polyphenols. Extensive research has shown that green tea polyphenols exhibit anti-inflammatory, antioxidative, and anti-microbial attributes, preventing or alleviating many terminal diseases, such as cardiovascular disease, neurodegenerative diseases, and osteoporosis.

**Figure 4 biomolecules-12-00371-f004:**
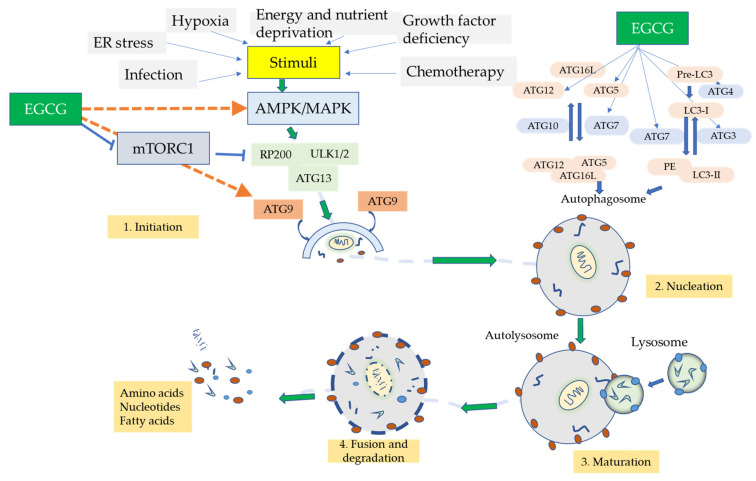
Proposed mechanism of EGCG autophagic response. Autophagy is a molecular recycling system explored extensively in cancer research. The promotion of molecular insults, i.e., ER stress, hypoxia, and chemotherapy, initiate the AMPK or MAPK/AKT/mTOR pathway, which recruits downstream effector protein kinases Unc-51 like autophagy activating kinase (ULK1/2,) and RP200 leading to the cascade of events leading to degradation of cellular debris. EGCG can act via AMPK or MTORC1 and act indirectly on autophagic-like proteins (ATG9) and ATG-9-15 to induce the autophagic series of events resulting in cellular debris breakdown and leading to the residue being reused for proper cellular growth and homeostasis.

**Figure 5 biomolecules-12-00371-f005:**
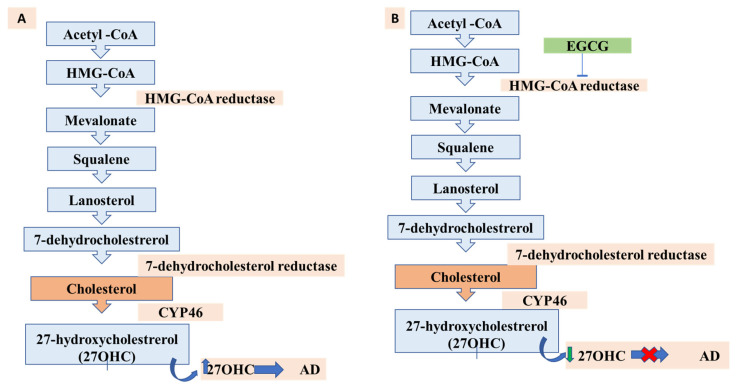
Contributing factors of cholesterol biosynthesis to AD genesis and possible inhibition by EGCG. (**A**) the normal cascade of events involves converting acetyl coenzyme A (Acetyl CoA) to 3-hydroxy-3-methylglutaryl-CoA by hydroxymethylglutaryl-CoA (HMG-CoA) synthase, which is changed to mevalonate by HMG-CoA reductase. A succession of enzymatic reactions converts mevalonate into 3-isoprenyl pyrophosphate, farnesyl pyrophosphate, squalene, lanosterol, and cholesterol leading to the generation of the oxysterol 27-hydroxycholesterol due to the enzymatic action of cholesterol 24S-hydroxylase (CYP46). The high amounts of oxysterols, i.e., 27 hydroxycholesterols (27-OHC the role), can contribute to AD biogenesis. (**B**) EGCG acting similar to a statin may be able to negate this elevation by inhibiting the regulative enzyme HMG-COA reductase, which will reduce the production of the oxysterols by preventing the enzymatic conversion of mevalonate resulting in the prevention of AD.

**Figure 6 biomolecules-12-00371-f006:**
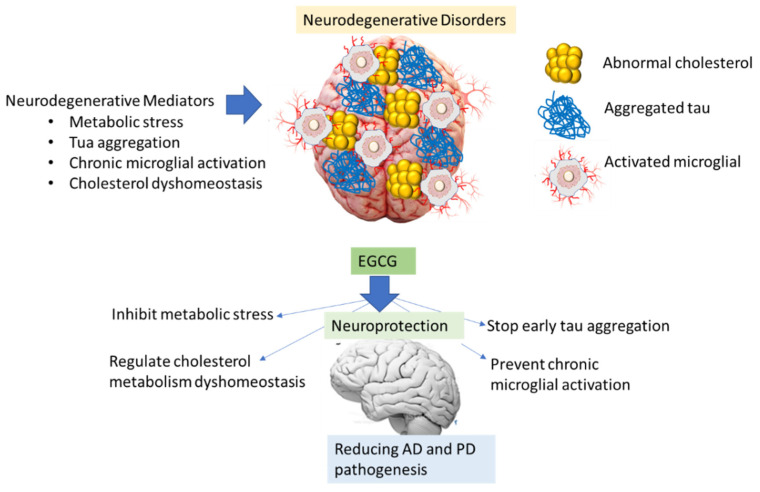
Picture Summary of EGCG highlights of this paper. Picture summary of key medicinal actions of EGCG as discussed in this review article.

**Table 1 biomolecules-12-00371-t001:** Possible mechanisms of EGCG neuroprotective action in AD demonstrated in various disease models.

Pharmacological Action	Model (In Vivo/In Vitro)	Therapeutic Focus	Reference
Anti-inflammatory	In vivo transgenic mice, human volunteers, Elegans, hamsters, Drosophila	A review of the role of green tea (*Camellia sinensis*) in antiphotoaging, stress resistance, neuroprotection, and autophagy	[161]
	In vitro and in vivo airway epithelial cells, experimental autoimmune encephalomyelitis (EAE), rat aortas, RA human patients, human umbilical vein endothelial cells, and synovial fibroblasts	The prevention and treatment of vascular inflammation in rheumatoid arthritis	[185]
	In vivo: human	Molecular pharmacology of inflammation: medicinal plants as anti-inflammatory agents	[170]
	In vitro: adult human ventricular cardiomyocyte cell line AC16	Suppresses cigarette smoking-induced inflammation in human cardiomyocytes, utilizing ROS-mediated MAPK and NF-κB pathways	[171]
	In vivo: Wistar albino female rats	Attenuates arthritis by regulating Nrf2, HO-1, and cytokine levels in an experimental arthritis model	[175]
	In vivo: Human mast cell line	Blockade of RANKL/RANK signaling pathway by EGCG alleviates mast cell-mediated inflammatory reactions	[174]
Antioxidant/oxidative stress	In vitro: human bronchial epithelial cells	Diminishes cigarette smoke-induced oxidative stress, lipid peroxidation, and inflammation in human bronchial epithelial cells	[172]
	In vitro: Single ventricular myocytes	Protective effects against lead-induced oxidative damage	[177]
	In vivo: Adult male Wistar albino rats	Attenuates cadmium-induced chronic renal injury and fibrosis	[179]
	multiple human and cell model systems	EGCG Management of Heavy Metal-Induced Oxidative Stress: Mechanisms of Action, Efficacy, and Concerns	[149]
	In vitro BEAS-2B cells and EBV-BL cells	Protects against chromate-induced toxicity in vitro	[178]
	In vivo Male 14-day-old Wistar rats	Protects rat brain mitochondria against cadmium-induced damage	[176]
Amyloid-beta formation inhibition	In vivo human adult brain	Microglial dysfunction in brain aging and Alzheimer’s disease	[65]
	In vitro N2a-APP695 cells	EGCG attenuates β-amyloid generation and oxidative stress involvement of PPARγ	[196]
	In vitro and in vivo Human embryonic kidney cells human CNS-derived neuroblastoma cells, APPHEK293 and SwAPP-N2a cells. Biomarkers of CSF, plasma, blood-derived brain exosomes, monocytes, and peripheral blood mononuclear cells	Autophagy modulation as a treatment of amyloid diseases	[197]
	In vitro hippocampal neurons, mouse, rat brain cells, and yeast model overexpressing htt	EGCG Amyloid Aggregation and Neurodegenerative Diseases	[158]
	In vivo mixed POPC/POPG (7:3) lipid bilayer	Plays a dual role in Aβ 42 protofibril disruption and membrane protection: A dynamic molecular study	[198]
	In vitro Synthetic Aβ42, Aβ fibril formulation	The effect of (−)-epigallocatechin-3-gallate on the amyloid-β secondary structure	[199]
Autophagy	In vitro C3H10T1/2 cells and 3T3-L1 preadipocytes	Effects of EGCG on autophagic lipolysis in adipocytes	[166]
	In vitro HEPG2 cells	A new molecular mechanism underlying the EGCG mediated autophagic modulation of AFP	[147]
	Human endothelial cells	Protects vascular endothelial cells from oxidative stress induced damage by targeting the autophagy dependent PI3K-Akt-mTOR pathway	[167]
	HEK293T cells	Promotes autophagy dependent survival via influencing the balance of mTOR -AMPK pathways upon endoplasmic reticulum stress	[168]
Cholesterol/Lipid Metabolism	high-fat diet induced mouse obesity model, human volunteers, rats,	The beneficial effects of principal polyphenols from green tea, coffee, wine, and curry on obesity	[37]
	BV2 cells and Twenty-four-week-old male C57BL/6J mice	Attenuates neuroinflammation in palmitic acid-stimulated bv-2 microglia and high-fat diet-induced obese mice	[200]
	C57BL/6 mice	prevents inflammation and diabetes -induced glucose tolerance through inhibition of NLRP3 inflammasome activation	[201]
	broiler chickens	Effects of EGCG on lipid metabolism and its underlying molecular mechanism	[202]
Anti-Aging	SH-SY5Y cells, SAMP10, and ddy mice	A review of the role of green tea (*Camellia sinensis*) in antiphotoaging, stress resistance, neuroprotection, and autophagy	[161]
	36 weeks old, spontaneously hypertensive rats and male normotensive Wistar-Kyoto rats	Green tea suppresses brain aging	[203]
	3T3-L1 preadipocytes	Cerebral cortex apoptosis in early aged hypertension: effects of epigallocatechin-3-gallate	[204]
	Wistar albino rats	Suppresses premature senescence of preadipocytes by inhibition of PI3K/Akt/mTOR pathway and induces senescent cell death by regulation of Bax/Bcl-2 pathway	[205]
MicroRNA	APP/PS1 transgenic mouse model	Identification of circulating mir-125b as a potential biomarker of Alzheimer’s disease	[193]
	chondrocytes, human THP-1 monocytic cells, and primary human fibroblasts	Quercetin, epigallocatechin gallate, curcumin, and resveratrol: from dietary sources to human microrna modulation	[192]
	Sprague Dawley Rats, chondrocytes, monocytes, and mice	Targeting miRNAs by polyphenols: a novel therapeutic strategy for aging	[191]

**Table 2 biomolecules-12-00371-t002:** Current clinical studies of EGCG on AD and PD.

Study Identifier	Study Type	Study Population	Study Purpose	Study Participants	Number of Patients Recruited	Intervention	Status
NCT03978052	Interventional	Spain	Prevention of cognitive decline in ApoE4 carriers with subjective cognitive decline after EGCG and a multimodal intervention	Alzheimer’s Disease cognitive function nutritional intervention	200	Dietary Supplement: EGCG, Placebo EGCG, personalized intervention, and lifestyle recommendations	Recruiting
NCT00951834	Interventional	Germany	Sunphenon EGCg (Epigallocatechin-Gallate) in the early stage of Alzheimer’s disease	Alzheimer’s Disease	21	Drug: Epigallocatechin-Gallate Drug Placebo	Completed
NCT00461942	Interventional	China	Efficacy and safety of green tea polyphenol in de novo Parkinson’s disease patients	Parkinson’s Disease	480	Drug: Green Tea Polyphenols (EGCG/ECG)	Completed

## Data Availability

Not applicable.

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
