# Peer review of "Epigallocatechin-3-Gallate (EGCG): New Therapeutic Perspectives for Neuroprotection, Aging, and Neuroinflammation for the Modern Age"

_biomolecules, 2022, doi:10.3390/biom12030371_

Round 1

Reviewer 1 Report

The authors made an updated review on AD as well as the neuroprotective effect of EGCG in AD. The review is detailed and with abundant content. They reviewed the pathogenesis of AD including cellular and molecular factors, important molecules and pathways involved in AD, phytochemical use in AD, the neuroprotective effect of EGCG and possible mechanisms from in vitro and in vivo models. This review provided a reference for the potential phytochemical treatment for AD.

However, the manuscript is quite dispersive and not well organized. The content for AD is too tedious which can be refined and some parts can be summarized. The authors should emphasize the application and mechanisms of the neuroprotective effect of EGCG. The following issues should be considered and revised before publication.

  1. The title and related sections should be re-organized more logically.
  2. In the abstract, the authors mentioned “Green tea polyphenols that have displayed neuroprotective action is epigallocatechin-3-gallate (EGCG)” (line 17), EGCG isn’t the only green tea polyphenol that displayed neuroprotective.
  3. Almost all of Part 2 introduced contents about AD while the title of this part is neurodegeneration. Thus, the title of Part 2 changes to AD is more suitable.
  4. The authors sometimes use AD (line 77), while sometimes uses Alzheimer's disease (line 84), which makes the contents disordered.
  5. Some full term was missed, “early onset AD” for “EOAD” (line 85-86), “LDL” (line 99), “CNS” (line 134), while other abbreviation gave repetitive in the same paragraph, “MicroRNAs (miRNAs)” (line 719)
  6. Figure 1 makes the etiological factors of AD more confusing. It can be clearer. Does age relate with neuronal death? What does the green line refer to?
  7. Part 4 can be summarized which is less related with title of this review.
  8. For Line 558 “EGCG, which was shown…”, the author mentioned the potential role of polyphenol then suddenly introduced EGCG and its role in nervous system. Rather more summary about the potential role of polyphenol is needed. And the role of EGCG in this part can be mentioned in the following sections.
  9. In part 5.1, please also briefly include the neuroprotective role of other compounds, in addition to EGCG in green tea.
  10. For Line 722 “Resveratrol…”, this part introduced EGCG and AD, please justify the importance of this part.
  11. For table1, please distinguish in vitro and in vivo models in the chart.
  12. The conclusion should emphasis the neuroprotective role of egcg and protentional mechanisms and applications in AD which make a summary of the review.
  13. Some statements should add references, such as line62, line 344-346 “Current research …”, lin 647-649 “Green tea…”, line 708-710 “In cardiovascular disease…”
  14. The manuscript needs re-editing. For example: Line 715 (MicroRNAs tiny…) is lack of predicate.

Author Response

Reviewer 1 Comments and Suggestions for Authors

The authors made an updated review on AD as well as the neuroprotective effect of EGCG in AD. The review is detailed and with abundant content. They reviewed the pathogenesis of AD including cellular and molecular factors, important molecules and pathways involved in AD, phytochemical use in AD, the neuroprotective effect of EGCG and possible mechanisms from in vitro and in vivo models. This review provided a reference for the potential phytochemical treatment for AD.

However, the manuscript is quite dispersive and not well organized. The content for AD is too tedious which can be refined and some parts can be summarized. The authors should emphasize the application and mechanisms of the neuroprotective effect of EGCG. The following issues should be considered and revised before publication.

  1. The title and related sections should be re-organized more logically.

Response: The title was changed Epigallocatechin-3-Gallate (EGCG: New Therapeutic Perspectives for Neuroprotection, Aging and Neuroinflammation for the Modern Age reflect a more current prospectus of this review article. The whole paper was rearranged and new sections were included (clinical analysis, in vitro/in vivo studies and other related analyses).

  1. In the abstract, the authors mentioned “Green tea polyphenols that have displayed neuroprotective action is epigallocatechin-3-gallate (EGCG)” (line 17), EGCG isn’t the only green tea polyphenol that displayed neuroprotective.

Author Response; Abstract has been changed from:

Abstract:  The exact etiology of Alzheimer’s disease (AD) is not well known; however, environmental, molecular, and genetic influences play a major role in the pathogenesis of this disease. AD's pathological findings include the aggregation of Amyloid Beta (Aβ) peptides, mitochondrial dysfunction, synaptic degradation caused by inflammation, elevated reactive oxygen species (ROS), and cerebrovascular dysregulation. Other aspects of AD progression are the dysregulated clearance of Aβ and blood-brain barrier (BBB) instability. It is now widely accepted that clustered activated microglia are hallmarks of the inflammatory pathology in AD. Meanwhile, phytochemicals or nutraceuticals have shown promise in modulating AD progression due to their roles as antioxidants, anti-inflammatory, and iron chelator action. Green tea polyphenols that have displayed neuroprotective action is epigallocatechin-3-gallate (EGCG), which has shown many therapeutic properties such as acting as a free radical scavenger and inhibiting the formation of Aβ. The anti-inflammatory, anti-aging, antioxidative, and cholesterol modulating properties of EGCG enable it to be an effective preventive treatment for AD. This review is focused on the chemical and pharmacological properties of EGCG involved in reducing oxidative stress, mitochondrial dysfunction, neuroinflammation, autophagy, and other factors associated with AD.

Modified version

Abstract: Alzheimer’s and Parkinson’s diseases are the two most common forms of neurodegenerative diseases. The exact etiology of these disorders is not well known; however, environmental, molecular, and genetic influences play a major role in the pathogenesis of these diseases. Using Alzheimer’s disease (AD) as the archetype, the pathological findings include the aggregation of Amyloid Beta (Aβ) peptides, mitochondrial dysfunction, synaptic degradation caused by inflammation, elevated reactive oxygen species (ROS), and cerebrovascular dysregulation. This review highlights the neuroinflammatory and neuroprotective role of Epigallocatechin 3-Gallate (EGCG): the medicinal component of Green Tea, a known nutraceutical that has shown promise in modulating AD progression, due to its role as antioxidant, anti-inflammatory, and anti-aging abilities. This report also reexamines the current literature and provides innovative approaches that EGCG can be used a s a preventive measure to alleviate AD and other neurodegenerative disorders.

  1. Almost all of Part 2 introduced contents about AD while the title of this part is neurodegeneration. Thus, the title of Part 2 changes to AD is more suitable.

Author’s Response: Section 2 Neurodegeneration was changed to Neurodegeneration: Examined in two prevalent neuroregressive disorders in which both Alzheimer’s and Parkinson’s are discussed in tandem. They are used as the archetypes for neurodegenerative disease throughout this review article.

  1. The authors sometimes use AD (line 77), while sometimes uses Alzheimer's disease (line 84), which makes the contents disordered.

Author’s rebuttal: The paragraph in question used the abbreviation AD to represent Alzheimer’s disease, which was defined in the heading. AD consistency was followed throughout the paper.

  1. Some full term was missed, “early onset AD” for “EOAD” (line 85-86), “LDL” (line 99), “CNS” (line 134), while other abbreviation gave repetitive in the same paragraph, “MicroRNAs (miRNAs)” (line 719)

Author’s Response: Early onset Alzheimer’s disease (EOAD) was introduced and discussed along with late onset alzheimer’s disease (LOAD).  Time was spent thoroughly reviewing the article to make sure that all of the abbreviations were properly defined.

  1. Figure 1 makes the etiological factors of AD more confusing. It can be clearer. Does age relate with neuronal death? What does the green line refer to?

Author’s Response: The figure has been revised to show the connection of protein aggregation and misfolding to neuronal death.  Age and family history can serve as contributors to AD in regards to genetics and age of onset.  This is discussed in great detail in the section entitled Alzheimer’s disease, where LOAD and EOAD are mentioned.

  1. Part 4 can be summarized which is less related with title of this review.

Author’s Response: This section was summarized and compressed into Contributing determinants to the incidence of AD: Health Disparities, Sex, and Gender

  1. For Line 558 “EGCG, which was shown…”, the author mentioned the potential role of polyphenol then suddenly introduced EGCG and its role in nervous system. Rather more summary about the potential role of polyphenol is needed. And the role of EGCG in this part can be mentioned in the following sections.

Author’s Response: This section was modified to discuss polyphenol usage in neurodegenerative disease. New sections were added to address the bioavailability of EGCG as well as to discuss its usage in other related inflammatory and age related disorders.

  1. In part 5.1, please also briefly include the neuroprotective role of other compounds, in addition to EGCG in green tea.

Author’s Response: Response; A section on green tea derivatives was added. Gallic acid was discussed in great detail; related to neurodegenerative disease as well as epicatechin (EC).

  1. For Line 722 “Resveratrol…”, this part introduced EGCG and AD, please justify the importance of this part.

Author’s Response; This section was modified under section Role of microRNA (miRNA) in neuroinflammation: Medicinal action by EGCG

  1. For table1, please distinguish in vitro and in vivo models in the chart.

Author’s Response: In vivo/vitro designation was added.

  1. The conclusion should emphasis the neuroprotective role of egcg and protentional mechanisms and applications in AD which make a summary of the review.

Author’s Response: A summary picture was created.

  1. Some statements should add references, such as line62, line 344-346 “Current research …”, lin 647-649 “Green tea…”, line 708-710 “In cardiovascular disease…”

Author’s Response: References were included.

  1. The manuscript needs re-editing. For example: Line 715 (MicroRNAs tiny…) is lack of predicate.

Author’s Response: The manuscript was revised entirely. 

Reviewer 2 Report

An interesting topic has been chosen while the topic has been already reviewed many times in the decades.

  • Zhang S, Zhu Q, Chen JY, OuYang D, Lu JH. The pharmacological activity of epigallocatechin-3-gallate (EGCG) on Alzheimer's disease animal model: A systematic review. Phytomedicine. 2020;79:153316. doi: 10.1016/j.phymed.2020.153316. PMID: 32942205.
  • Fernandes L, Cardim-Pires TR, Foguel D, Palhano FL. Green Tea Polyphenol Epigallocatechin-Gallate in Amyloid Aggregation and Neurodegenerative Diseases. Front Neurosci. 2021 Sep 14;15:718188. doi: 10.3389/fnins.2021.718188. PMID: 34594185; PMCID: PMC8477582.
  • Gruendler R, Hippe B, Sendula Jengic V, Peterlin B, Haslberger AG. Nutraceutical Approaches of Autophagy and Neuroinflammation in Alzheimer's Disease: A Systematic Review. Molecules. 2020 18;25(24):6018. doi: 10.3390/molecules25246018. PMID: 33353228; PMCID: PMC7765980.
  • Cascella M, Bimonte S, Muzio MR, Schiavone V, Cuomo A. The efficacy of Epigallocatechin-3-gallate (green tea) in the treatment of Alzheimer's disease: an overview of pre-clinical studies and translational perspectives in clinical practice. Infect Agent Cancer. 2017;12:36. doi: 10.1186/s13027-017-0145-6. PMID: 28642806; PMCID: PMC5477123.

There are a few major issues that need to be addressed.

Page 1; Line 9-20: The background or introduction of the review in the abstract is too long. Need to be concise and highlighted mainly in context to EGCG. Further, there is a missing outcome of the review in terms of significant findings and future direction.

Page 2; Line 69: Required to add a section as Methods or data search immediately after the introduction which will justify the construction of the manuscript in a critical way. It should include the selection criteria such as the use of what kind of database sources were used (Such as Scopus, MEDLINE, PubMed, Cochrane, and ScienceDirect etc.), how did the filter occur (how many years of studies, and types of study including in vitro, in vivo and clinical studies). Further, if any clinical significance of the studies is available then needs to be highlighted in each subsection.

Page 14; Line 694: According to the Title of the manuscript, most of the sections are irrelevant and not directly related to Alzheimer’s disease, however, the main section 5.3 has been described very poorly and is limited.

Author Response

Reviewer 2 Comments and Suggestions for Authors

An interesting topic has been chosen while the topic has been already reviewed many times in the decades.

  • Zhang S, Zhu Q, Chen JY, OuYang D, Lu JH. The pharmacological activity of epigallocatechin-3-gallate (EGCG) on Alzheimer's disease animal model: A systematic review. Phytomedicine. 2020;79:153316. doi: 10.1016/j.phymed.2020.153316. PMID: 32942205.
  • Fernandes L, Cardim-Pires TR, Foguel D, Palhano FL. Green Tea Polyphenol Epigallocatechin-Gallate in Amyloid Aggregation and Neurodegenerative Diseases. Front Neurosci. 2021 Sep 14;15:718188. doi: 10.3389/fnins.2021.718188. PMID: 34594185; PMCID: PMC8477582.
  • Gruendler R, Hippe B, Sendula Jengic V, Peterlin B, Haslberger AG. Nutraceutical Approaches of Autophagy and Neuroinflammation in Alzheimer's Disease: A Systematic Review. Molecules. 2020 18;25(24):6018. doi: 10.3390/molecules25246018. PMID: 33353228; PMCID: PMC7765980.
  • Cascella M, Bimonte S, Muzio MR, Schiavone V, Cuomo A. The efficacy of Epigallocatechin-3-gallate (green tea) in the treatment of Alzheimer's disease: an overview of pre-clinical studies and translational perspectives in clinical practice. Infect Agent Cancer. 2017;12:36. doi: 10.1186/s13027-017-0145-6. PMID: 28642806; PMCID: PMC5477123.

There are a few major issues that need to be addressed.

Page 1; Line 9-20: The background or introduction of the review in the abstract is too long. Need to be concise and highlighted mainly in context to EGCG. Further, there is a missing outcome of the review in terms of significant findings and future direction.

Author Response; Abstract has been changed from:

Original

Abstract:  The exact etiology of Alzheimer’s disease (AD) is not well known; however, environmental, molecular, and genetic influences play a major role in the pathogenesis of this disease. AD's pathological findings include the aggregation of Amyloid Beta (Aβ) peptides, mitochondrial dysfunction, synaptic degradation caused by inflammation, elevated reactive oxygen species (ROS), and cerebrovascular dysregulation. Other aspects of AD progression are the dysregulated clearance of Aβ and blood-brain barrier (BBB) instability. It is now widely accepted that clustered activated microglia are hallmarks of the inflammatory pathology in AD. Meanwhile, phytochemicals or nutraceuticals have shown promise in modulating AD progression due to their roles as antioxidants, anti-inflammatory, and iron chelator action. Green tea polyphenols that have displayed neuroprotective action is epigallocatechin-3-gallate (EGCG), which has shown many therapeutic properties such as acting as a free radical scavenger and inhibiting the formation of Aβ. The anti-inflammatory, anti-aging, antioxidative, and cholesterol modulating properties of EGCG enable it to be an effective preventive treatment for AD. This review is focused on the chemical and pharmacological properties of EGCG involved in reducing oxidative stress, mitochondrial dysfunction, neuroinflammation, autophagy, and other factors associated with AD.

Modified version

Abstract: Alzheimer’s and Parkinson’s diseases are the two most common forms of neurodegenerative diseases. The exact etiology of these disorders is not well known; however, environmental, molecular, and genetic influences play a major role in the pathogenesis of these diseases. Using Alzheimer’s disease (AD) as the archetype, the pathological findings include the aggregation of Amyloid Beta (Aβ) peptides, mitochondrial dysfunction, synaptic degradation caused by inflammation, elevated reactive oxygen species (ROS), and cerebrovascular dysregulation. This review highlights the neuroinflammatory and neuroprotective role of Epigallocatechin 3-Gallate (EGCG): the medicinal component of Green Tea, a known nutraceutical that has shown promise in modulating AD progression, due to its role as antioxidant, anti-inflammatory, and anti-aging abilities. This report also reexamines the current literature and provides innovative approaches that EGCG can be used a s a preventive measure to alleviate AD and other neurodegenerative disorders.

Future Directions  has been modified from:

Original

AD is on pace to be the fastest-growing worldwide epidemic in the next few years [188]. Current research on AD should be focused on advancing our understanding of the role of inflammation, cholesterol metabolism, autophagy, and oxidative stress to create a more robust alternative therapy to combat AD. Metabolomic/ lipidomic approaches can be utilized to go beyond and understand the interconnections of metabolic disorders to AD progression and evolution. Further examination of EGCG’s effects on astrocytes and oligodendrocytes relative to AD will foster new Aβ clearance modulators and protein aggregation disruptors to be used. More research is needed on sex, gender, and health inequality effects on the immunological and epigenetic aspects of the CNS related to AD risk. A more in-depth understanding of aging and autophagy in glial cells is recommended

Revision:

AD is on pace to be the fastest-growing worldwide epidemic in the next few years [188]. Current research on AD should be focused on advancing our understanding phytochemical intervention in inflammaging, cholesterol metabolism, microglia-neuronal interactions, epigenetics, neuroprotection, and autophagy to create a more robust alternative therapy to combat AD. The enhancement of EGCG bioavailability and combination immunotherapies may be of scientific interest.  Research focused on BBB barrier dysfunction and mitochondrial disruption by environmental toxicants on in vitro models and the administration of Green tea catechins such as EC and EGCG. Scientific analysis on health disparities and comorbidities, especially in association with metabolic stress may be of concern. This review also highlights the need for in vitro studies on green tea catechins (EGCG/EC) coupled with physical activity and calorie restriction in preventing neurodegenerative disease. Lastly evaluate preventive measures by green tea catechins on protein aggregation/misfolding, and neuroinflammation in concussion and reduction of susceptibility to neurodegenerative pathogenesis.

Page 2; Line 69: Required to add a section as Methods or data search immediately after the introduction which will justify the construction of the manuscript in a critical way. It should include the selection criteria such as the use of what kind of database sources were used (Such as Scopus, MEDLINE, PubMed, Cochrane, and ScienceDirect etc.), how did the filter occur (how many years of studies, and types of study including in vitro, in vivo and clinical studies). Further, if any clinical significance of the studies is available then needs to be highlighted in each subsection.

Author’s Response: Throughout this study Pubmed, Scopus, and ScienceDirect were used and up to date articles were utilized throughout this review. A section discussing the current clinical studies of EGCG to PD and AD has been added.

Page 14; Line 694: According to the Title of the manuscript, most of the sections are irrelevant and not directly related to Alzheimer’s disease, however, the main section 5.3 has been described very poorly and is limited.

Author’s Response: The title has been changed to Epigallocatechin-3-Gallate (EGCG): New Therapeutic Perspectives for Neuroprotection, Aging and Neuroinflammation for the Modern Age, which better reflects the central aims of this paper. Section 5.3 Epigallocatechin-3-Gallate (EGCG) and Alzheimer’s disease has been modified to section 9.  Epigallocatechin-3-Gallate (EGCG) Therapeutic action in AD and PD

Reviewer 3 Report

This review is focused on the chemical and pharmacological properties of epigallocatechin-3-gallate (EGCG) involved in reducing oxidative stress, mitochondrial dysfunction, neuroinflammation, and apoptosis associated with Alzheimer’s Disease (AD). Please conduct the following concerns.

  1. Role of Tau-protein was not conducted in current submission. Why?
  2. In general, AD is known to be progressive. What is the best stage for EGCG?
  3. Oral intake of EGCG may pass the BBB or not that needs to indicate in clear.
  4. In addition to improvement of AD, did EGCG alleviate Parkinsonism? This is an interesting topic for readers.
  5. Autophagy by EGCG was mainly documented in peripheral tissues. Is it observed in patients or animals with AD?
  6. Merits of EGCG and/or tea have been identified in clinical practice?
  7. Variations of EGCG from chemical antioxidant(s) were ignored.
  8. Before conclusion, astrocytes and oligodendrocytes were speculated as the future target without clear reason(s).
  9. Conclusions show the weakness of current review. Please indicate the new data mentioned in current report in brief showing the novelty.

Author Response

Reviewer 3 Comments and Suggestions for Authors

This review is focused on the chemical and pharmacological properties of epigallocatechin-3-gallate (EGCG) involved in reducing oxidative stress, mitochondrial dysfunction, neuroinflammation, and apoptosis associated with Alzheimer’s Disease (AD). Please conduct the following concerns.

  1. Role of Tau-protein was not conducted in current submission. Why?

Author Response:  It was neglected, because the focus was on microglia and neuroinflammation relative to AD. Although all events related to AD progression occur simultaneously based on the literature it seems as though Tau post translational modifications related to neurofibrillary tangles occurs at a stage post Aβ oligomeric aggregative to which EGCG may prove ineffective.  There is also a dearth of information about tau aggregation and EGCG. Current research is just beginning to understand the relationship of tau to microglia.  I thought it would be better to relate EGCG to the traumatic brain injury, cardiovascular disease, and other comorbidities to heighten the awareness of EGCG to a varied audience.

  1. In general, AD is known to be progressive. What is the best stage for EGCG?

Author response: The early stages of AD progression primarily during the oligomeric period of Aβ formation.  This is evidenced in the following papers:

Gonçalves, P. B., Sodero, A., & Cordeiro, Y. (2021). Green Tea Epigallocatechin-3-gallate (EGCG) Targeting Protein Misfolding in Drug Discovery for Neurodegenerative Diseases. Biomolecules11(5), 767. https://doi.org/10.3390/biom11050767

Zhang S, Zhu Q, Chen JY, OuYang D, Lu JH. The pharmacological activity of epigallocatechin-3-gallate (EGCG) on Alzheimer's disease animal model: A systematic review. Phytomedicine. 2020;79:153316. doi: 10.1016/j.phymed.2020.153316. PMID: 32942205.

Fernandes L, Cardim-Pires TR, Foguel D, Palhano FL. Green Tea Polyphenol Epigallocatechin-Gallate in Amyloid Aggregation and Neurodegenerative Diseases. Front Neurosci. 2021 Sep 14;15:718188. doi: 10.3389/fnins.2021.718188. PMID: 34594185; PMCID: PMC8477582. Cascella M, Bimonte S, Muzio MR, Schiavone V, Cuomo A. The efficacy of Epigallocatechin-3-gallate (green tea) in the treatment of Alzheimer's disease: an overview of pre-clinical studies and translational perspectives in clinical practice. Infect Agent Cancer. 2017;12:36. doi: 10.1186/s13027-017-0145-6. PMID: 28642806; PMCID: PMC5477123

  1. Oral intake of EGCG may pass the BBB or not that needs to indicate in clear.

Author rebuttal: To address this issue a section on EGCG bioavailability was added.

  1. In addition to improvement of AD, did EGCG alleviate Parkinsonism? This is an interesting topic for readers.

Author’s Response: In the section entitled Epigallocatechin-3-Gallate (EGCG) neurodegenerative disease aggregation. Tau aggregation was halted and a possible mechanism of action was given.EGCG’s actions on parkinsonism based on current in vivo studies is also discussed.

  1. Autophagy by EGCG was mainly documented in peripheral tissues. Is it observed in patients or animals with AD?

Author’s Response: In vivo animal studies have mostly been investigated in cancer research. There is a dearth of knowledge related to neurodegenerative disease, however, an epigenetics study taken from Khalil, H entitled “Aging is associated with hypermethylation of autophagy genes in macrophages. This study showed that EGCG’s methylation inhibitor properties were able to inhibit DNA methyltransferase 2 (DNMT2) (which is correlated to increased levels of methylated autophagy molecules ATg5 and LC3B) in C57BL/6 aged (62-64 weeks old) mice. This is important because it further illustrates the relationship of EGCG and Tau aggregation.

  1. Merits of EGCG and/or tea have been identified in clinical practice?

Authors Response: A new table was added to address current clinical trials utilizing EGCG.  A section that discusses clinical studies of EGCG has been included.

  1. Variations of EGCG from chemical antioxidant(s) were ignored.

Author’s Response; The section entitled Green Tea and its derivatives: Therapeutic actions related to inflammation and neurodegeneration discusses the variations of EGCG. Gallic acid was discussed in great detail related to neurodegenerative disease as well as epicatechin (EC).

  1. Before conclusion, astrocytes and oligodendrocytes were speculated as the future target without clear reason(s).

Authors response: To answer this question I would advise reading the following article Castellani, G., & Schwartz, M. (2020). Immunological Features of Non-neuronal Brain Cells: Implications for Alzheimer's Disease Immunotherapy. Trends in immunology, 41(9), 794–804. https://doi.org/10.1016/j.it.2020.07.005. Astrocytes and oligodendrocytes along with schwann cells (collectively called glia) are linked to microglia but for the focus of this paper, the intention was focused on microglia.  However, the relationship of glia to neuroinflammation/neurodegeneration is still being researched.  The section Future Directions has been revised to address more relevant issues related to neurodegeneration, aging, and neuroinflammation.

Original Future Directions:

Original

AD is on pace to be the fastest-growing worldwide epidemic in the next few years [188]. Current research on AD should be focused on advancing our understanding of the role of inflammation, cholesterol metabolism, autophagy, and oxidative stress to create a more robust alternative therapy to combat AD. Metabolomic/ lipidomic approaches can be utilized to go beyond and understand the interconnections of metabolic disorders to AD progression and evolution. Further examination of EGCG’s effects on astrocytes and oligodendrocytes relative to AD will foster new Aβ clearance modulators and protein aggregation disruptors to be used. More research is needed on sex, gender, and health inequality effects on the immunological and epigenetic aspects of the CNS related to AD risk. A more in-depth understanding of aging and autophagy in glial cells is recommended

Revision:

AD is on pace to be the fastest-growing worldwide epidemic in the next few years [188]. Current research on AD should be focused on advancing our understanding phytochemical intervention in inflammaging, cholesterol metabolism, microglia-neuronal interactions, epigenetics, neuroprotection, and autophagy to create a more robust alternative therapy to combat AD. The enhancement of EGCG bioavailability and combination immunotherapies may be of scientific interest.  Research focused on BBB barrier dysfunction and mitochondrial disruption by environmental toxicants on in vitro models and the administration of Green tea catechins such as EC and EGCG. Scientific analysis on health disparities and comorbidities, especially in association with metabolic stress may be of concern. This review also highlights the need for in vitro studies on green tea catechins (EGCG/EC) coupled with physical activity and calorie restriction in preventing neurodegenerative disease. Lastly evaluate preventive measures by green tea catechins on protein aggregation/misfolding, and neuroinflammation in concussion and reduction of susceptibility to neurodegenerative pathogenesis.

  1. Conclusions show the weakness of current review. Please indicate the new data mentioned in current report in brief showing the novelty.

Author’s Response: Conclusion has been changed from:

Neurodegenerative diseases remain a complex issue that will require multidisciplinary and multifaceted ways to investigate. Recent advances in research have shown that natural products may offer an alternative remedy to solving this problem. It is also noteworthy to discuss the innovations in our knowledge of the mechanisms involved in AD pathogenesis: aging, environmental toxicants, genetics, and inflammation. The myriad medicinal uses of nutraceuticals, specifically EGCG, may offer a new path to reducing the future implications of neurological diseases.

To:

This review showcased the therapeutic actions of polyphenols and introduced the medical benefits of green tea catechins.  The ameliorative and neuroprotective properties of EGCG were discussed in relation to neuroinflammation, aging, protein aggregation, and autophagy (Figure 6).  EGCG was shown to quell neuroinflammation by reducing microglial activation.  Aging was discussed as the main factor of heightening neurodegenerative disease development.  This was discussed in association with the immunosenescence of microglia.  AD and PD were used as the main archetypes of neurodegenerative pathology and both are rising in significance as global aging populations increase.  The protein Tau was introduced and discussed for its role in understanding fibrilogenesis. Autophagy a common research interest in cancer has gained interest in neurodegenerative disease to understand the dysregulated clearance mechanisms demonstrated in PD and AD. Metabolic stress was examined in relation to the current curative properties of EC and various anti-oxidative functions of GA were deliberated about. The postulated role of EGCG in mediating cholesterol metabolism was introduced.  Lastly a debated topic of health disparities, sex and gender was included to deal with the challenge of unequal access to curative therapies due to socioeconomic conditions. EGCG remains a promising therapeutic strategy in the battle against neurodegenerative disease.

Round 2

Reviewer 1 Report

The paper is well revised and good for publication.